# Activation of the dopaminergic pathway from VTA to the medial olfactory tubercle generates odor-preference and reward

Zhijian Zhang[1,2†], Qing Liu[1†], Pengjie Wen[1], Jiaozhen Zhang[1], Xiaoping Rao[1], Ziming Zhou[3], Hongruo Zhang[3], Xiaobin He[1], Juan Li[1], Zheng Zhou[4], Xiaoran Xu[3], Xueyi Zhang[3], Rui Luo[3], Guanghui Lv[2], Haohong Li[2], Pei Cao[1], Liping Wang[4], Fuqiang Xu[1,2]*

[1]Center for Brain Science, Key Laboratory of Magnetic Resonance in Biological Systems, Wuhan Institute of Physics and Mathematics, Chinese Academy of Sciences, Wuhan, China; [2]Wuhan National Laboratory for Optoelectronics, Wuhan, China; [3]College of Life Sciences, Wuhan University, Wuhan, China; [4]Shenzhen Key Lab of Neuropsychiatric Modulation and Collaborative Innovation Center for Brain Science, CAS Center for Excellence in Brain Science, Shenzhen Institutes of Advanced Technology, Chinese Academy of Sciences, Shenzhen, China

**Abstract** Odor-preferences are usually influenced by life experiences. However, the neural circuit mechanisms remain unclear. The medial olfactory tubercle (mOT) is involved in both reward and olfaction, whereas the ventral tegmental area (VTA) dopaminergic (DAergic) neurons are considered to be engaged in reward and motivation. Here, we found that the VTA (DAergic)-mOT pathway could be activated by different types of naturalistic rewards as well as odors in DAT-cre mice. Optogenetic activation of the VTA-mOT DAergic fibers was able to elicit preferences for space, location and neutral odor, while pharmacological blockade of the dopamine receptors in the mOT fully prevented the odor-preference formation. Furthermore, inactivation of the mOT-projecting VTA DAergic neurons eliminated the previously formed odor-preference and strongly affected the Go-no go learning efficiency. In summary, our results revealed that the VTA (DAergic)-mOT pathway mediates a variety of naturalistic reward processes and different types of preferences including odor-preference in mice.
DOI: https://doi.org/10.7554/eLife.25423.001

*For correspondence:
fuqiang.xu@wipm.ac.cn

[†]These authors contributed equally to this work

**Competing interests:** The authors declare that no competing interests exist.

## Introduction

Sensory perceptions are influenced by life experiences. For us, it's a pleasure to unexpectedly smell the odor of grandmother's cookies that we loved in our childhood, and people of different cultures may have different odor-preferences, such as blue cheese to European and stinky bean-curd to Chinese. For rodents, reward-related odor perception is critical for food foraging and reproduction (*Doty, 1986*). The formation of odor-preference should involve the interaction between the olfactory and reward systems, although the mechanisms, the engaged brain regions and neural circuits are still unclear.

The olfactory tubercle (OT) is a long and narrow tubular brain structure located at the ventral part of striatum. It belongs to the olfactory system and receives dense direct inputs from the main olfactory bulb (OB) and other olfactory cortexes such as the anterior olfactory nucleus (AON) and piriform cortex (PCX) (*Wesson and Wilson, 2011*). As a part of the ventral striatum reward circuitry, the OT is also heavily innervated by dopaminergic (DAergic) neurons from the ventral tegmental area (VTA, [*Voorn et al., 1986*]). Belonging to both the olfactory and reward circuits (*Ikemoto, 2007*;

*Wesson and Wilson, 2011*), the OT is at the proper position to play critical roles in the formation of odor-preference. Previous results from relatively limited studies are in agreement with this intuition. The OT encodes natural reinforcers (*Gadziola and Wesson, 2016*), odor valence and pheromone reward (*Gadziola and Wesson, 2016*; *DiBenedictis et al., 2015*; *Agustín-Pavón et al., 2014*), and also influences reward behaviors and odor-preferences (*Fitzgerald et al., 2014*). Furthermore, specific domains of the OT represent reward or punishment-associated odor-cues (*Murata et al., 2015*). These works have revealed that the OT is involved in the formation of odor-preference. The OT directly receives olfactory inputs; however, how it is regulated by the reward system to form odor-preferences and whether it plays roles in other reward behaviors are still largely unknown.

An earlier study showed that the VTA can modulate the activities of olfactory-related OT neurons (*Mooney et al., 1987*). The VTA DAergic neurons have been proposed to play central roles in general reward and motivation, via innervating the ventral striatum (*Hu, 2016*; *Roeper, 2013*; *Ungless et al., 2010*). As a component of the ventral striatum, most of the OT neurons are D1 and D2 dopamine receptor-expressing medium spiny neurons (*Murata et al., 2015*; *Millhouse and Heimer, 1984*), which are heavily innervated by DAergic neurons from the VTA (*Voorn et al., 1986*). From these anatomical and functional facts about the two brain regions, we hypothesized that the VTA (DAergic)-OT pathway play pivotal roles in the formation of odor-preference and reward behaviors involving other modalities. Compared with the lateral OT, the medial OT (mOT) is more involved in addiction, reward and reward associated odor perceptions (*Murata et al., 2015*; *Ikemoto, 2005*; *Ikemoto, 2003*). In fact, the mOT mediates self-administration of cocaine or D-amphetamine even more robustly than the nucleus accumbens (NAc), which is crucially important for reward learning and drug abuse (*Koob and Volkow, 2010*; *Ikemoto, 2005*, *2003*; *Rodd-Henricks et al., 2002*). Therefore, we focused on the pathway made up of VTA DAergic projection and the mOT (VTA (DAergic)-mOT) to test our hypothesis that it plays a key role in the formation of odor-preference and the other reward behaviors.

In attempting to answer these questions, we first confirmed the direct DAergic projection from VTA to the mOT by using virus-based tracing system. Using fiber photometry recording, we subsequently showed that the activities of the VTA-mOT DAergic projection and the mOT can be changed by reward or odor stimulation respectively. Besides, optogenetic activation of the VTA (DAergic)-mOT pathway was able to activate multiple brain regions and form space/location preferences (even overcame the aversion to central zone). Further, coupling this activation with odor stimulation simultaneously could not only generate preference for a neutral odor, but also abolished the avoidance for an aversive odor. On the other hand, inactivating the mOT-projecting VTA DAergic neurons, or blocking the dopamine receptors in the mOT, abolished the expression of previously formed odor-preferences. These results demonstrate that activation of the VTA (DAergic)-mOT pathway generates odor-preference as well as the other types of preferences. Thus, this pathway might be a key player in modulating perception of a given stimulus.

## Results

### Mapping the input network of the mOT

To map the direct inputs of the mOT, a recombinant RV expressing green florescent protein (RV-dG-GFP) was injected into the mOT (*Figure 1A,B*). RV-labeled neurons were widely distributed in the olfactory systems including the OB, PCX (*Figure 1C–F*), and entorhinal cortex (data not shown), as well as reward related brain regions such as a large area of the VTA (*Figure 1G*). In the OB, a large number of neurons in the mitral cell layers as well as a small number of neurons in the external plexiform layer were labeled by RV (*Figure 1D,E*). Besides, RV also labeled many neurons in other brain regions including the hippocampus and dorsal raphe (data not shown). Our findings are consistent with previous studies that the mOT is connected with both olfactory and reward systems (*Ikemoto, 2007*; *Wesson and Wilson, 2011*), but with more certainty by excluding the possibility of labeling the passing-by fibers.

We then examined the cell types of RV infected neurons in the VTA by immunochemical staining of the tyrosine hydroxylase (TH), a biomarker of DAergic neurons, and found about half of the GFP labeled cells (55.32%, n = 3 mice) were TH positive (*Figure 1H–J*).

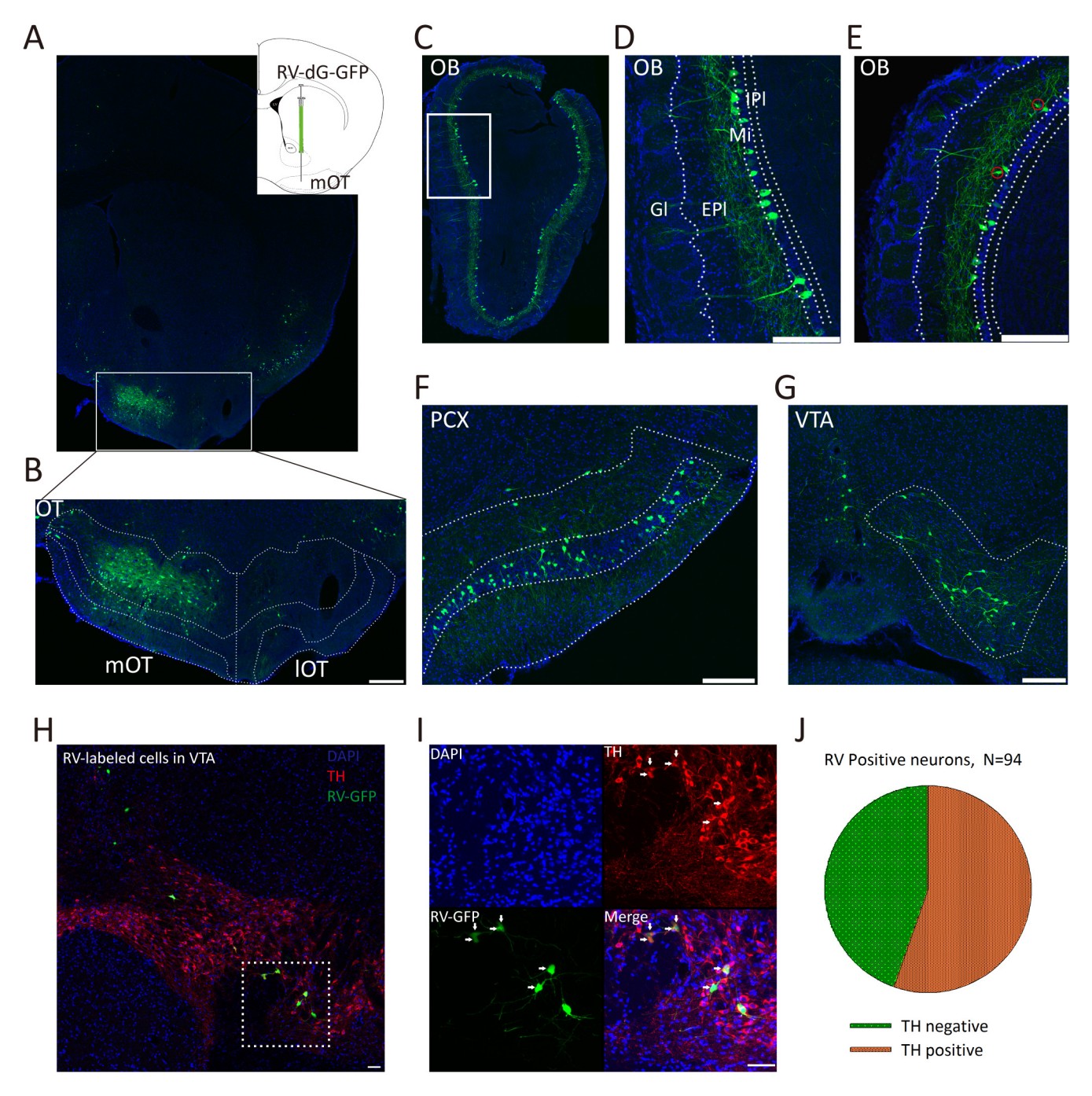

**Figure 1.** Inputs of the mOT. (**A**) Schematic for the tracing experiment. RV-dG-GFP was unilaterally injected into the mOT of C57BL/6 mice. The representative coronal brain section showed the injection site in the mOT. (**B**) Magnification of RV-dG-GFP labeled neurons in the mOT. (**C–G**) GFP positive cells were widely distributed in the OB (**C–E**), PCX (**F**) and VTA (**G**) with RV retrograde tracing. In the OB, the labeled cells were mainly distributed in the mitral cell layers (**D**), while also sparsely located in the external plexiform layer (E, red circles). (**H, I**) TH was co-stained with RV-dG-GFP labeled neurons in the VTA, followed by mOT injection of RV-dG-GFP. (**J**) Statistical chart showed that more than half of the GFP + neurons co-express TH (n = 3). Nuclei of all the slices were stained blue with DAPI. Scale bar: 200 μm.

DOI: https://doi.org/10.7554/eLife.25423.002

The following source data is available for figure 1:

**Source data 1.** Statistical result of RV infected neurons in the VTA that co-stained with tyrosine hydroxylase (TH).

DOI: https://doi.org/10.7554/eLife.25423.003

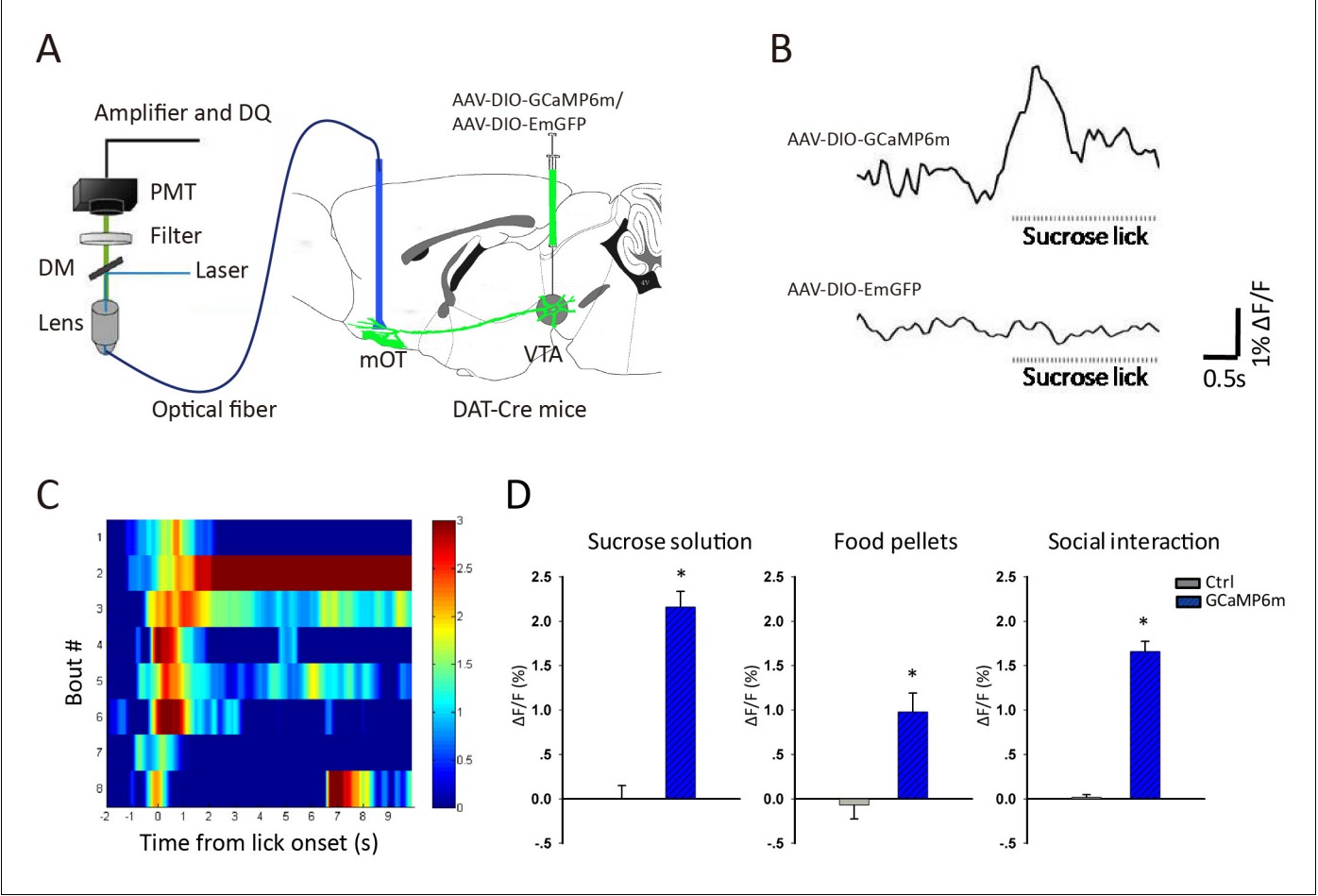

**Figure 2.** Reward increased Ca$^{2+}$ signals of VTA-mOT DAergic projection. (**A**) Schematic of the fiber photometry setup for recording of the VTA-mOT DAergic projection from DAT-Cre mice. (**B**) Raw traces of GCaMP and control GFP fluorescence as the mice lick sucrose solution. ΔF/F represents change in fluorescence from the mean level before task. (**C**) The heatmap illustrated Ca$^{2+}$ signal responses of eight sucrose licking bouts from a representative mouse. Color bar: ΔF/F. (**D**) Responses to sucrose solution (left), food pellets (middle) and social interaction (right) for the control and GCaMP6m-expressing mice. *p<0.05.

DOI: https://doi.org/10.7554/eLife.25423.004

The following source data and figure supplements are available for figure 2:

**Source data 1.** Fiber photometry recording ΔF/F value to sucrose solution, food pellets and social interaction for the control and GCaMP6m-expressing mice.

DOI: https://doi.org/10.7554/eLife.25423.007

**Figure supplement 1.** The VTA-mOT DAergic projection showed a higher response to sucrose reward than the bitter quinine solution.

DOI: https://doi.org/10.7554/eLife.25423.005

**Figure supplement 2.** The mOT neurons were activated by different odor stimulations.

DOI: https://doi.org/10.7554/eLife.25423.006

## The VTA-mOT DAergic projection responded to naturalistic reward

Both the OT neurons and the VTA DAergic neurons play roles in reward processing (*Gadziola et al., 2015*; *Hu, 2016*). Although the mOT is densely innervated by the VTA DAergic neurons, the specific functions of this pathway in rewarding have not been tested in vivo. Thus, we used calcium-dependent fiber photometry (*Gunaydin et al., 2014*) to detect the VTA-mOT DAergic projection responding to naturalistic reward stimulations in free moving mice.

Energy is the basic requirement for the survival of animals, so we first detected the response of this pathway in water- or food-deprived mice to sucrose solution and solid food pellet, respectively. For this purpose, we infused AAV-DIO-GCaMP6m into the VTA of DAT-Cre mice, and recorded the

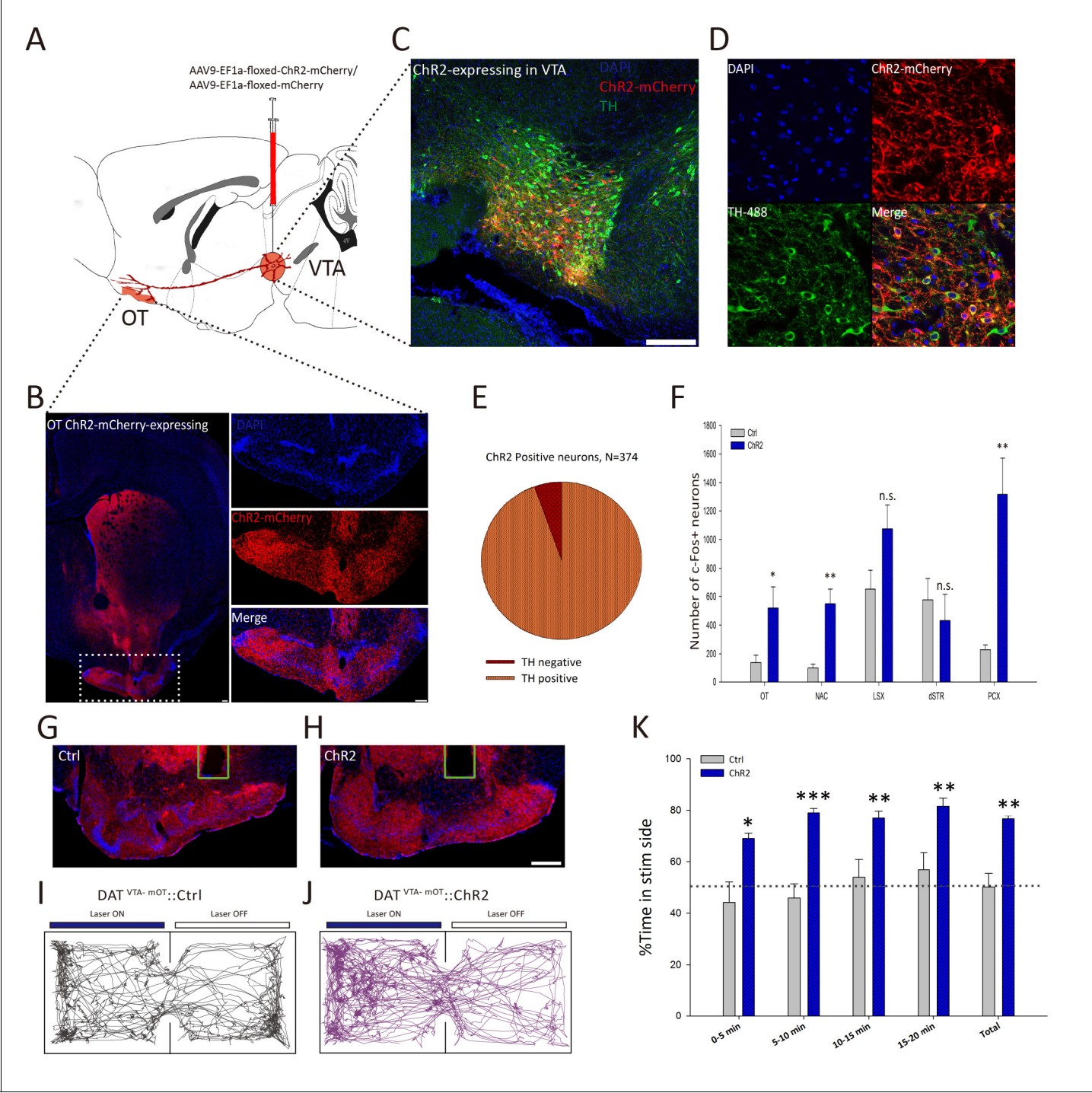

**Figure 3.** Activation of the VTA-mOT DAergic projection produced acute place preference. (**A**) Schematic for virus injection to manipulate the VTA-mOT DAergic projections. AAV9-EF1a-floxed-ChR2-mCherry or the AAV9-EF1a-floxed-mCherry was unilaterally injected into the VTA of DAT-Cre mice. (**B**) ChR2-mCherry positive axons were densely distributed in the mOT. (**C, D**) Co-localization of the TH with ChR2-mCherry in the VTA neurons. (**E**) Statistical chart showed that ChR2-mCherry was relatively restricted to TH-positive neurons (n = 3). (**F**) Blue light stimulation of the VTA-mOT DAergic projections increased the c-fos expression levels in the OT, NAc, and PCX, while the c-fos expression levels in the LSX and dSTR had no significant changes (n.s., p>0.05). (**G, H**) Optical fibers were implanted in the mOT of control or ChR2-expressing mice. (**I, J**) Representative real-time place preference (RTPP) tracks illustrated light-evoked behavioral preference in ChR2-expressing mice (right, n = 7), but not in control mice (left, n = 7). (**K**) ChR2-expressing mice showed a significant preference for light stimulation chamber compared with control mice in the RTPP test. *p<0.05, **p<0.01, ***p<0.001. Scale bar: 200 μm.

DOI: https://doi.org/10.7554/eLife.25423.008

*Figure 3 continued on next page*

*Figure 3 continued*

The following source data and figure supplement are available for figure 3:

**Source data 1.** Statistical result of c-fos-positive cells in the OT, NAc, LSX, dSTR and PCX after blue light stimulation of the VTA-mOT DAergic projections.
DOI: https://doi.org/10.7554/eLife.25423.010
**Figure supplement 1.** Optogenetic activation of the VTA-mOT DAergic projections.
DOI: https://doi.org/10.7554/eLife.25423.009

fluorescence signals of the VTA-mOT DAergic terminals (*Figure 2A*). When the mice sought and acquired sucrose solution, the GCaMP fluorescence was reliably increased (p<0.001, t-test, n = 6, *Figure 2B–D*-left). Sweetened water mainly provides a taste reward, while solid food is a reward related to multiple sensory systems. We then tested the solid food intake and found that the GCaMP signals were also reliably increased with food pellet consumption (p<0.001, t-test, n = 5, *Figure 2D*-middle).

In addition, social interaction is critical for animals, we then investigated whether the VTA-mOT DAergic projection is involved. We examined the effect of male-female and male-male interactions, and observed reliable increases of GCaMP fluorescence individually. Because no significant difference of fluorescence changes were observed between the male-female and male-male social interactions, we pooled the data of social interaction experiments (p<0.001, t-test, n = 6, *Figure 2D*-right). For control mice in all the fiber photometry experiments (n = 3), only small random signals were observed (*Figure 2B,D*).

Since freely behaving mice refused to consume bitter food or water, to compare the response of the VTA-mOT DAergic projection between reward and aversive stimulus, the sucrose or quinine solutions were orally delivered to head-fixed mice directly through a metal cannula. We found that the VTA-mOT DAergic projection showed a higher response to reward stimuli compared with aversive ones (*Figure 2—figure supplement 1*).

## The mOT activated by odor stimulation

Since the OT has been considered as a component of the olfactory system and the mOT receives direct inputs from the OB (*Figure 1C–E*), we next tested whether the mOT responds to different kinds of odor stimulations. To verify that the mOT activities can be activated by odor stimulations, we infused AAV-hsyn-GCaMP6s into the mOT of C57BL/6J mice. Calcium signals of the mOT neurons evoked by different odor stimulations were recorded from head-fixed mice (*Figure 2—figure supplement 2A*). It was found that the mOT neurons were generally activated by different odor stimulations, including isoamyl acetate (ISO), carvone (CAR), citral (CIT), and geraniol (GER), respectively (*Figure 2—figure supplement 2B*). The results demonstrated that the mOT was indeed involved in olfactory functions, which are both consistent with and complement to previous studies using immunohistological and electrophysiological methods (*Gadziola et al., 2015*; *Carlson et al., 2014*), irrespective of the odor types.

## Opto-stimulation of the VTA-mOT DAergic projection elicited reward behaviors

We showed that the VTA-mOT DAergic projection is activated by diverse naturalistic rewards, but the behavioral effects elicited by the activation of this pathway are still unknown. To selectively manipulate this pathway, we injected AAV9-EF1a-floxed-ChR2-mCherry into the VTA of DAT-Cre transgenic mice (*Figure 3A*). Six weeks after viral infection, we observed mCherry expression in the somas of the VTA neurons (*Figure 3—figure supplement 1A*, bottom), as well as dense mCherry-positive axon fibers in the mOT (*Figure 3B*). The negative mCherry signal in the VTA of C57BL/6 mice (*Figure 3—figure supplement 1A*, top) with the same injection indicated that ChR2-mCherry would only express in Cre-positive DAergic neurons in DAT-Cre mice. To further verify the specificity of ChR2-mCherry expression in the DAergic neurons of the VTA, we quantified the number of the VTA neurons that were TH-positive (TH+) and mCherry-positive (mCherry+). We found that about 94.39% of the mCherry+ neurons were also labeled with TH (*Figure 3C–E*), suggesting that the expression of ChR2-mCherry is largely restricted to DAergic neurons. Whole cell recording of light-evoked activity from ChR2-expressing cells in brain slices from these mice revealed that 20 Hz blue

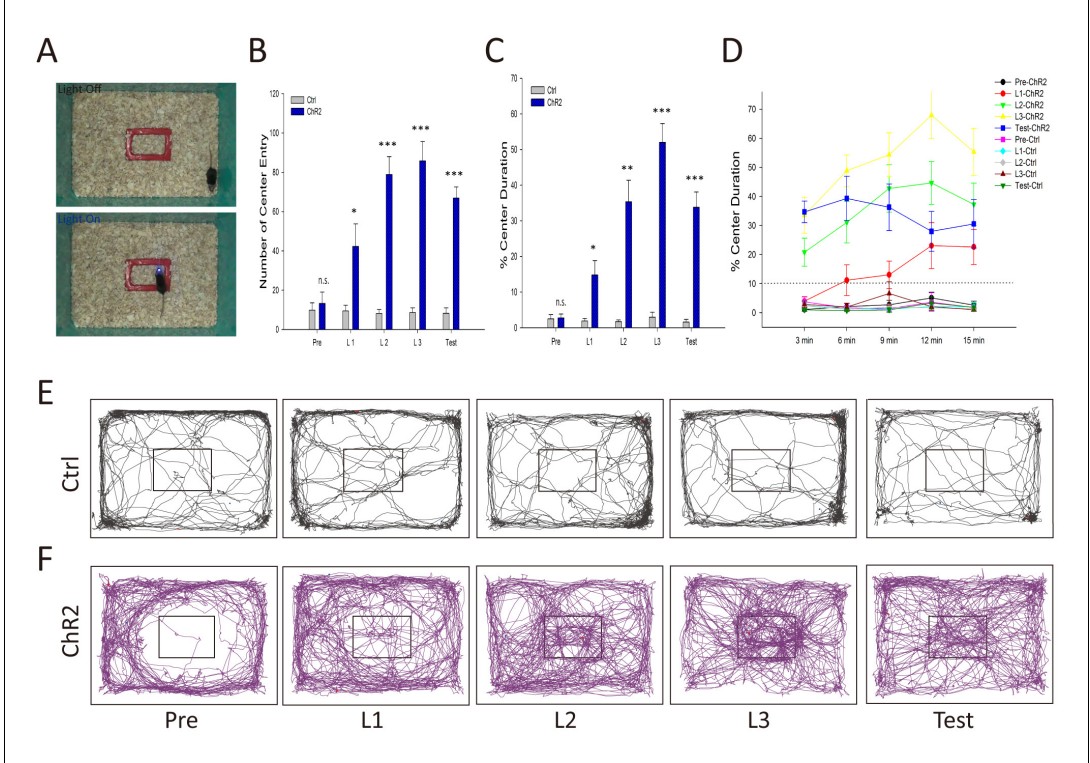

**Figure 4.** Activation of the VTA-mOT DAergic projection produced positive reinforcement. (**A**) Schematics for iClass training. Mice explored in an open field containing a rectangle in the center, which is marked red and 1/10 of the total area, for 5 consecutive days. In the first day (Pre) and last day (Test) mice received no light stimulation. During the 3 consecutive training days (**L1, L2, L3**), light pulses (483 nm) at 20 Hz were delivered to activate the VTA-mOT DAergic projection when mice were in the rectangle (bottom), and terminated until they left the rectangle (Top). (**B–D**) ChR2-mCherry-expressing mice (n = 7) increased entry times (**B**) and retention (**C, D**) for central area when compared with control mice (n = 7). The dashed line represents the percentage of the marked rectangle to the total. *p<0.05, **p<0.01, ***p<0.001. The preference was turned up at the first training day, increased with the training sections, and maintained to the test day (**B, C**). Learning curves showed that mice took 3–6 min to overcome their instinctive avoidance for the central area in the first training day (**D**). The center preference increased with training time and days. (**E, F**) Representative iClass tracks illustrated light-evoked center preference in ChR2-expressing mice (bottom, (**F**) but not in the control mice (top, (**E**).
DOI: https://doi.org/10.7554/eLife.25423.011

The following source data is available for figure 4:

**Source data 1.** Statistical result of the time percentage spent in the stimulated side in RTPP for the control and ChR2-expressing mice.
DOI: https://doi.org/10.7554/eLife.25423.012

light pulses reliably induced action potentials in ChR2-expressing neurons (*Figure 3*-fugure supplement 1B). These results laid the basis for further optogenetic manipulation of the DAergic VTA-mOT projection.

We then implanted optic fibers at the mOT to activate the VTA-mOT DAergic projection (*Figure 3G,H*). To reveal the brain regions responding to the activation of the pathway, we screened the c-fos expression (*Bepari et al., 2012*; *Herrera and Robertson, 1996*; *Worley et al., 1993*) in brain regions of the olfactory cortexes and reward-related areas. As expected, the OT showed a robust enhancement of c-fos activation (*Figure 3F*, *Figure 3—figure supplement 1C*. p=0.040, t-test), suggesting that it was strongly activated by stimulation of this pathway. In addition, the c-fos expression was significantly increased in the NAc and PCX compared to the control mice (*Figure 3F*, *Figure 3—figure supplement 1D*. p=0.006 for NAc, p=0.006 for PCX, t-test). The dorsal striatum (dSTR) and the lateral septal complex (LSX) exhibited a decreased and increased tendency of c-fos expression, respectively, although neither was significant (both p>0.05, t-test). These results indicated that both the reward and the olfactory systems were affected by the activation of the VTA-mOT DAergic fibers.

After we confirmed that the VTA-mOT DAergic projection and the mOT are involved in rewards as well as olfaction, and activating of this pathway affects both the reward and the olfactory system, we tested the hypothesis that activation of this pathway might produce odor and even other preferences. For this purpose, we first activated the VTA-mOT DAergic projection to test whether it was able to generate reward behaviors. Real-time place preference (RTPP) tests revealed that mice expressing ChR2 exhibited significant preference for the light-paired chamber (76.55 ± 1.25%, n = 7), compared with the control mice (50.22 ± 5.27%, n = 7) which only expressed mCherry (p=0.001, t-test, *Figure 3I–K*). These data suggested that acute activation of the VTA-mOT DAergic projection produces rewarding effects.

Activation of the VTA-mOT DAergic projection induced acute place preference, is this rewarding effect strong enough to overcome the instinctive avoidance and drive positive reinforcement? Tests with intra-Cranial light administration in a specific subarea (iClass, *Figure 4A*) exhibited that optic activation of mice expressing ChR2-mCherry (n = 7) dramatically increased center exploration, represented by the significantly enhanced entry and retention times, immediately after the first day of iClass training using 20 Hz light pulses (*Figure 4B–F*, p=0.017 for center entry times, p=0.017 for center duration in L1; *Videos 1–5*, t-test). We segmented the 15 min training course each day into every 3 min sections and found that mice rapidly overcame their instinctive avoidance for the central area in about 6 min from light activation in L1 (*Figure 4D*). The preference of mice for the light-paired central area increased with light training (2.76 ± 1.08, 14.81 ± 4.00, 35.34 ± 6.04 and 52.03 ± 5.27 central entry times; 13.29 ± 5.76%, 42.29 ± 11.49%, 78.86 ± 9.11%, and 85.86 ± 9.79% central retention for Pre, L1, L2 and L3, respectively) and was still maintained 24 hr later when no light pulse was delivered (33.80 ± 4.23 for central entry times, p<0.001; 66.86 ± 5.66% for central retention, p<0.001, t-test). No such effects were observed in the control mice (*Figure 4B–E*, all p>0.05, n = 7, t-test). The data suggest that activation of the VTA-mOT DAergic projection is able to overcome the instinctive avoidance and drive-positive reinforcement.

## Opto-stimulation of the VTA (DAergic)-mOT pathway coupled with odor exposure elicited odor-preference

We then tested whether activation of the VTA-mOT DAergic projection influences odor-preference. After 2 days of simultaneous pairing of optical stimulation in the mOT with odor stimulation, mice were tested for preference between the two odors on the following day without light activation (*Figure 5A,B*). As shown in *Figure 5C*, we chose one pair of odors (carvone and geraniol), each of them is equivalently sampled by naive mice and found that the ChR2-expressing mice showed a higher percentage of investigation time to the paired odor (geraniol) compared with the non-paired (carvone) (p=0.012, n = 8, t-test, *Video 6*) or the control mice (p=0.053, n = 8, t-test), while the two odors still remained equivalently sampled by the control mice after pairing with optical stimulation (p=0.575, n = 8, t-test). These results indicated that activation of the VTA-mOT DAergic projection can elicit odor-preference.

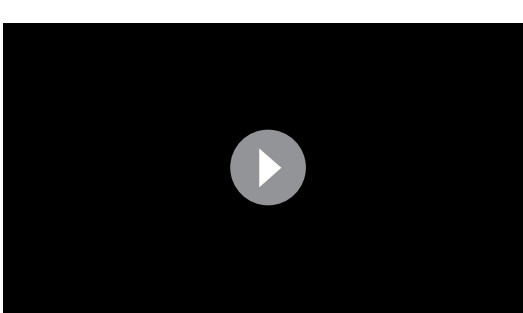

**Video 1.** Pre-phase (pre) of iClass task of a ChR2-expressing mouse. The fiber patch cord was connected but no light was delivered.
DOI: https://doi.org/10.7554/eLife.25423.013

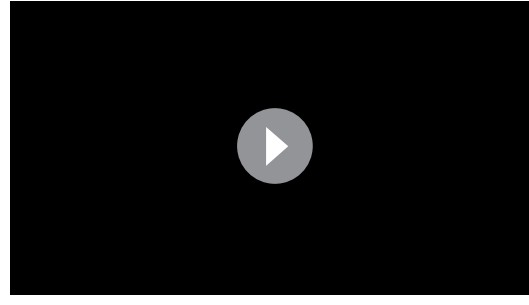

**Video 2.** The first light-training session (L1) of iClass task of the ChR2-expressing mouse. When the central of the animal was located within the marked central rectangular area, 10 mW blue light pulses at 20 Hz were constantly delivered into the mOT.
DOI: https://doi.org/10.7554/eLife.25423.014

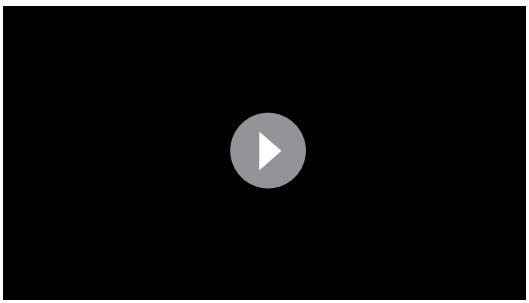

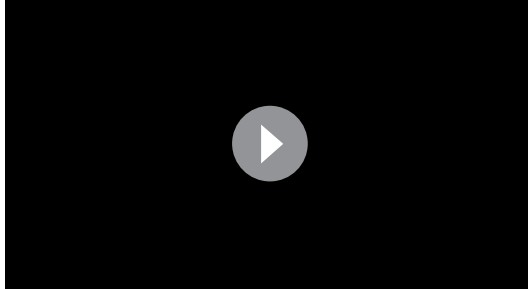

**Video 3.** The second light-training session (L2) of iClass task of the ChR2-expressing mouse. When the central of the animal was located within the marked central rectangular area, 10 mW blue light pulses at 20 Hz were constantly delivered into the mOT.
DOI: https://doi.org/10.7554/eLife.25423.015

**Video 4.** The third light-training session (L3) of iClass task of the ChR2-expressing mouse. When the central of the animal was located within the marked central rectangular area, 10 mW blue light pulses at 20 Hz were constantly delivered into the mOT.
DOI: https://doi.org/10.7554/eLife.25423.016

As the carvone and geraniol are equivalently sampled by mice inherently, we further tested whether it is possible to abolish or reverse the instinctive aversion to an odor when pairing it with this projection stimulation. We chose two other odors, a preferred (sesame butter) and an aversive (TMT) odor to mice (*Kobayakawa et al., 2007*). After 2 days of optical stimulation in the mOT pairing with TMT sampling, mice were tested on the following day without light activation. As shown in *Figure 5D*, after pairing, the ChR2-expressing mice (n = 8) spent a significantly higher percentage of investigation time for TMT, compared with the control mice (p=0.049, n = 8, t-test) and unpaired odor (p=0.025, t-test), reaching the level equivalent to sesame butter (p=0.680, n = 8). These results indicate that activation of the VTA-mOT DAergic projection can also abolish the avoidance of mice to an innately aversive odor.

However, optical stimulation of the VTA-mOT DAergic projection might lead to the activation of the passing-by DAergic fibers in the mOT or VTA DAergic cell bodies, which may then activate afferents projecting to the other brain regions. To exclude these possibilities and address the role of the local mOT dopamine release in producing the odor-preference, we infused the D1 and D2 dopamine receptor antagonists (SCH-23390 and raclopride) 5 min before pairing the projection activation with odor stimulation (*Figure 5E*). We found that administration of these drugs abolished the formation of preference for the paired odor (*Figure 5F*, p=0.757, n = 6, t-test), which is preferred by animals with only saline infusion (*Figure 5F*, p=0.027, n = 6, t-test). These results indicated that the local mOT dopamine released by the VTA (DAergic)-mOT pathway is responsible for the formation of odor-preference as we have observed.

We have shown that it is sufficient to generate odor-preference by activating the VTA (DAergic)-mOT pathway while exposed to the odor simultaneously, and that dopamine in the mOT is necessary for this process. However, the potentially supra-physiological activation produced by optogenetic manipulation might elicit artificial behaviors. Therefore, whether this pathway is really necessary for the formation of odor-preference under the influence of hedonic life experience still remains unclear. Further, whether or how this pathway plays roles in the perception of the preferred odors after the formation of odor-preference is also unknown. To address these questions, we decided to train water deprived mice for Go-no go task, with one of the two odors associated with water, a naturalistic reward, then examine the effects caused by the chemogenetical inactivation of the mOT-projecting VTA DAergic neurons. At first, we co-injected RV-dG-GFP into the NAc and RV-dG-dsRed into the mOT (*Figure 6A,B*) and found that the co-labeled rate (1.65%) of the VTA neurons were negligible, which implied that it may not be common for a VTA DAergic neuron to co-innervate the NAc and the mOT (*Figure 6C–E*). Thus, we are able to selectively target the mOT-projecting VTA DAergic neurons and further express hM4Di in these neurons (*Figure 7A,C*) to silence them. To achieve this goal, we injected CAV-CMV-Cre and AAV9-EF1a-floxed-hM4Di-mCherry or AAV9-EF1a-floxed-mCherry into the mOT and VTA, respectively, in C57BL/6 mice (*Figure 7A*). The mCherry positive cells are restricted to the VTA area (*Figure 7B,C*), and the projections in the OT are mainly distributed in the medial part (*Figure 7D*). We found that although the mCherry-positive

cells only took a small portion (37.20%) of the whole VTA DAergic neurons (**Figure 7—figure supplement 1A,B**), most (90.86%) of the mCherry-labeled cells were also co-expressing TH (**Figure 7E, F**). The ratio of co-labeled TH+ neurons produced by CAV is much higher than by RV-infection (**Figure 1J**). Since the expression of endogenous proteins are often severely reduced followed by RV infection, the lower co-labeling ratio for RV might be underestimated. Thus, with CAV-CMV-Cre retrograde transport from the mOT, we are able to selectively silence the mOT-projecting VTA DAergic neurons. We found that after the Go-no go training was performed, inactivating the mOT-projecting VTA DAergic neurons disrupted the conditioned odor-preference (**Figure 7G**), but intact odor discrimination remained (**Figure 7H**, left and middle bar pairs). These results indicated that the projection might be necessary for established odor-preference but not critical for odor discrimination. We further found that when the mOT-projecting VTA neurons were inactivated during the Go-no go training, the learning efficiency of the well-trained mice to a new odor pair was strongly affected (**Figure 7H**, right bar pair, 66.17% accuracy vs. 91.00% for control, with random level at 50%, p=0.002), suggesting the important roles of the projection to establish the paradigm which can lead to the formation of odor-preference.

Moreover, we found that dense axonal fibers were also distributed in the NAc for some mice, which was not consistent with the RV-labeled results (**Figure 6**). This might be caused by the large amount of CAV-CMV-Cre injected into the mOT, which may diffuse into the nearby regions. To exclude the possibility, we injected the AAV9-EF1a-floxed-eNpHR3.0-mCherry into the VTA of DAT-Cre mice and bilaterally inactivated the VTA-mOT DAergic projection in an odor-food-associative learning test (**Figure 8A–C**). The control mice showed a significant preference to the S+ odor after the learning session (**Figure 8D**, 52.43 ± 5.32% for pre learning, 67.78 ± 3.40% for post learning, p=0.018). However, optogenetic inhibiting the VTA-mOT DAergic projection during the odor-food associative learning disrupted the conditioned odor-preference of the eNpHR-expressing mice (**Figure 8D**, 51.82 ± 2.66% for pre learning, 57.07 ± 3.06% for post learning, p=0.134).

Together, these results indicated that the VTA-mOT DAergic projection should play important roles in the formation of odor-preference under the influence of hedonic life experience.

## Discussion

The OT has been implicated in both reward and olfactory information processing (**Wesson and Wilson, 2011**). In the present study, we first showed that various types of odors and diverse naturalistic reward stimulations can activate the mOT and the VTA-mOT DAergic projection, respectively. Then we found that activation of the VTA (DAergic)-mOT pathway can generate preference for a neutral odor or abolish the avoidance for an aversive odor; inhibition of the projection downstream in the mOT through specific dopamine receptor antagonists prevented odor-preference formation completely; and inactivation of the mOT-projecting VTA DAergic neurons fully blocked the formed odor-preference, and strongly attenuated the Go-no go learning performance which leads to the odor-preference established through naturalistic reward odor-preference. These data suggest that the VTA (DAergic)-mOT pathway mediates the formation of odor-preference.

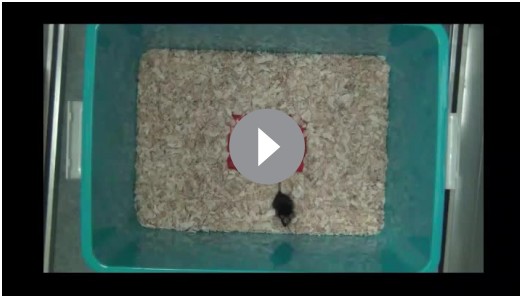

**Video 5.** The test day (test) of iClass task of the ChR2-expressing mouse. The fiber patch cord was connected but no light was delivered.
DOI: https://doi.org/10.7554/eLife.25423.017

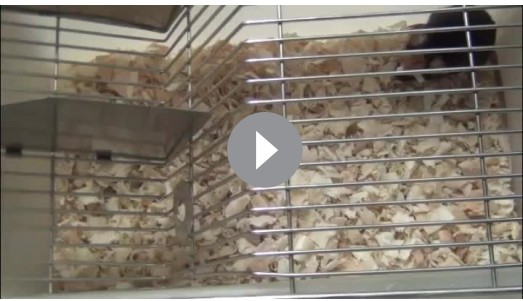

**Video 6.** Odor-preference test after odor-light pairing of a ChR2 expressing mouse.
DOI: https://doi.org/10.7554/eLife.25423.020

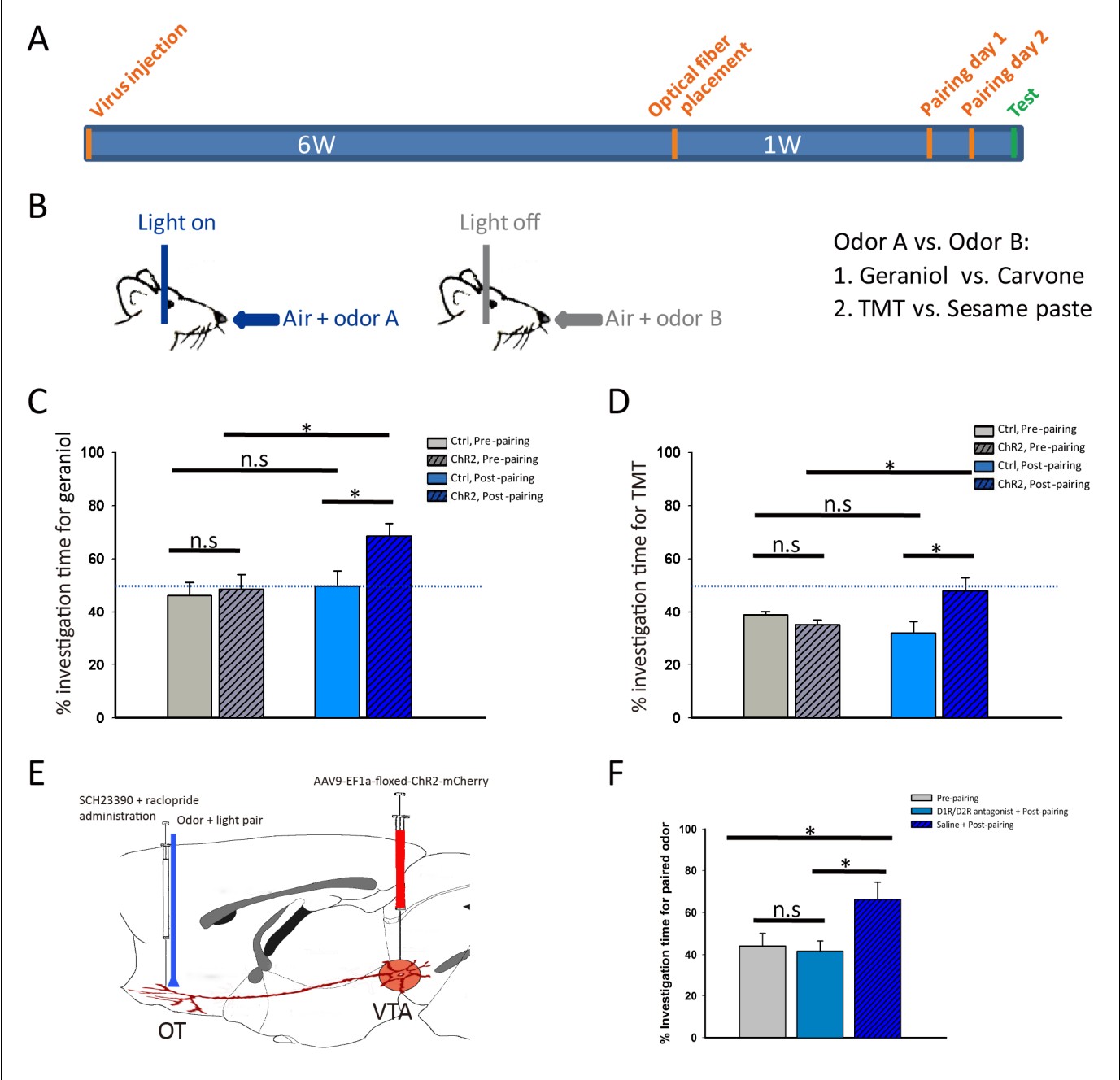

**Figure 5.** Activation of the VTA (DAergic)-mOT pathway elicited or altered odor-preference. (**A**) The time-course for optogenetic activation of the VTA (DAergic)-mOT pathway paired with odor stimulations and test for odor-preference. (**B**) Schematic for the odor-light pairing training. (**C**) For a pair of neutral odors which are equivalently sampled by the naive mice, the percentage of investigation time for the paired odor, geraniol, was significantly higher than the non-paired one, carvone after pairing with activation of this pathway (*p<0.05, n = 8). (**D**) For an odor pair with one that is preferred and one that is aversive to the naive mice, the percentage of investigation time was significantly increased for the aversive odor, TMT, equivalent to the previous preferred odor, sesame butter after pairing with activation of this pathway (*p<0.05, n = 8). (**E**) Schematic for pre-treatment of the D1R and D2R antagonist (SCH-23390 and raclopride) before optogenetic activation of the VTA (DAergic)-mOT pathway paired with odor stimulation. (**F**) The percentage of investigation time for the paired odor, geraniol, was not significantly changed followed by the antagonist administration (p=0.757, n = 6). However, the mice that pretreated with normal saline still showed significant preference to the paired odor (*p<0.05, n = 6).

DOI: https://doi.org/10.7554/eLife.25423.018

The following source data is available for figure 5:

**Source data 1.** Statistical result of center entries and percentage of center duration in iClass training for the control and ChR2-expressing mice.

DOI: https://doi.org/10.7554/eLife.25423.019

## VTA-mOT DAergic projection in reward

Although previous study has shown that electrical stimulation of the OT promotes reward (*Fitzgerald et al., 2014*), the behavioral effects have never been evaluated by directly targeting the VTA-mOT DAergic projection. It is reported that the ventral striatum is heterogeneously innervated by GABAergic, glutamatergic and DAergic afferents from the VTA. DAergic neurons in the VTA are involved in diverse functions, depending on their projection targets (*Kim et al., 2016*; *Cohen et al., 2012*). The roles of the VTA-mOT DAergic projection still remain unclear. Here, using optogenetics, we found that, activation of the VTA-mOT DAergic projection elicited RTPP (*Figure 3G–K*) (*Stamatakis and Stuber, 2012*), and the rewarding effect was strong enough to rapidly overcome the instinctive avoidance of mice to drive-positive reinforcement (*Figure 4*). Moreover, taking advantage of in vivo calcium-dependent fiber photometry, we found for the first time that the VTA-mOT DAergic projection was activated by diverse naturalistic rewards, including sucrose solution, solid food and social interactions (*Figure 2*). Our results consolidate the rewarding role of the mOT and demonstrate that the VTA-mOT DAergic projection is involved in this function.

Lacking cellular specificity, electrical stimulation might activate local neuron population, and axon fibers projecting to the area and passing-by non-selectively. Therefore, it was difficult to decide which component(s), local neurons, afferent to or efferent from the OT, and fibers passing-by the OT, contribute to the observed effects. Our results using optogenetics proved that activation of the VTA-mOT DAergic projection contributes to the rewarding role of the mOT, although the downstream neurons in the mOT involved in this pathway still need to be elucidated. It was reported that activation of the trimeric G-protein Gs-coupled D1R increases the excitability of the D1R + neurons, and that activation of the Gi-coupled D2R decreases the excitability of the D2R + neurons (*Stoof and Kebabian, 1981*). Thus, the rewarding effect observed here might be due to the simultaneous activation of D1R + neurons and inhibition of D2R + neurons followed by dopamine release in the mOT. Besides, cholinergic neurons in the OT also respond to DAergic terminal activation (*Wieland et al., 2014*), which might also lead to reward, because rats easily learned to self-administer cholinergic receptor agonists carbachol in the NAc (*Ikemoto et al., 1998*). Together, these researches implicated that the rewarding effects produced by stimulation of the VTA-mOT DAergic projection might be the results of diverse activation and/or inactivation response of different populations of neurons in the mOT. Further studies using optogenetic and chemogenetic to manipulate cell-type-specific neurons in the mOT with corresponding Cre-line transgenic mice for the elucidation of the downstream pathway would be of great interest.

On one hand, artificial activation of the VTA-mOT DAergic terminals generates reward behavior; on the other hand, this pathway is activated by naturalistic rewards, including sucrose solution, solid food and social interactions. It is reported that the activities of the VTA DAergic projection are target-dependent in response to rewards and shock (*Kim et al., 2016*). For example, the projection to NAc has been shown to increase activity during reward and decrease activity during shock, while the projection to basolateral-amygdala displayed increased response to both reward and shock. Also, the projection to prefrontal cortex exhibited increased activity to shock but not reward. We found that the VTA-mOT DAergic projection is activated by different types of naturalistic rewards, thus suggesting that it might mediate generalized rewarding processes. Besides, the VTA-mOT DAergic projection is also slightly activated by the bitter quinine solution, although it is more weakly activated than by sucrose solution. Furthermore, although the NAc/mOT-projecting VTA DAergic neurons may hardly be completely labeled limited by the efficiency of RV infection, it could be speculated that the VTA DAergic neurons may not commonly innervate both the NAc and mOT depending on the very low co-labeled rate of VTA neurons by RV (*Figure 6*). These results added new information to the functional heterogeneity of VTA DAergic projections.

## VTA (DAergic)-mOT pathway in odor and other preferences

Reward-related odor perception is critical for survival and reproduction of animals (*Doty, 1986*). As recent studies have shown that the OT neurons encode odor valence by enhancing firing rates (*Gadziola et al., 2015*), and that rewarding odor cues preferentially activate the D1R + neurons in the mOT (*Murata et al., 2015*), the mOT seems to be involved in odor-preference. However, the mechanism of how the odor-preference is established under the influence of life experience remains unclear.

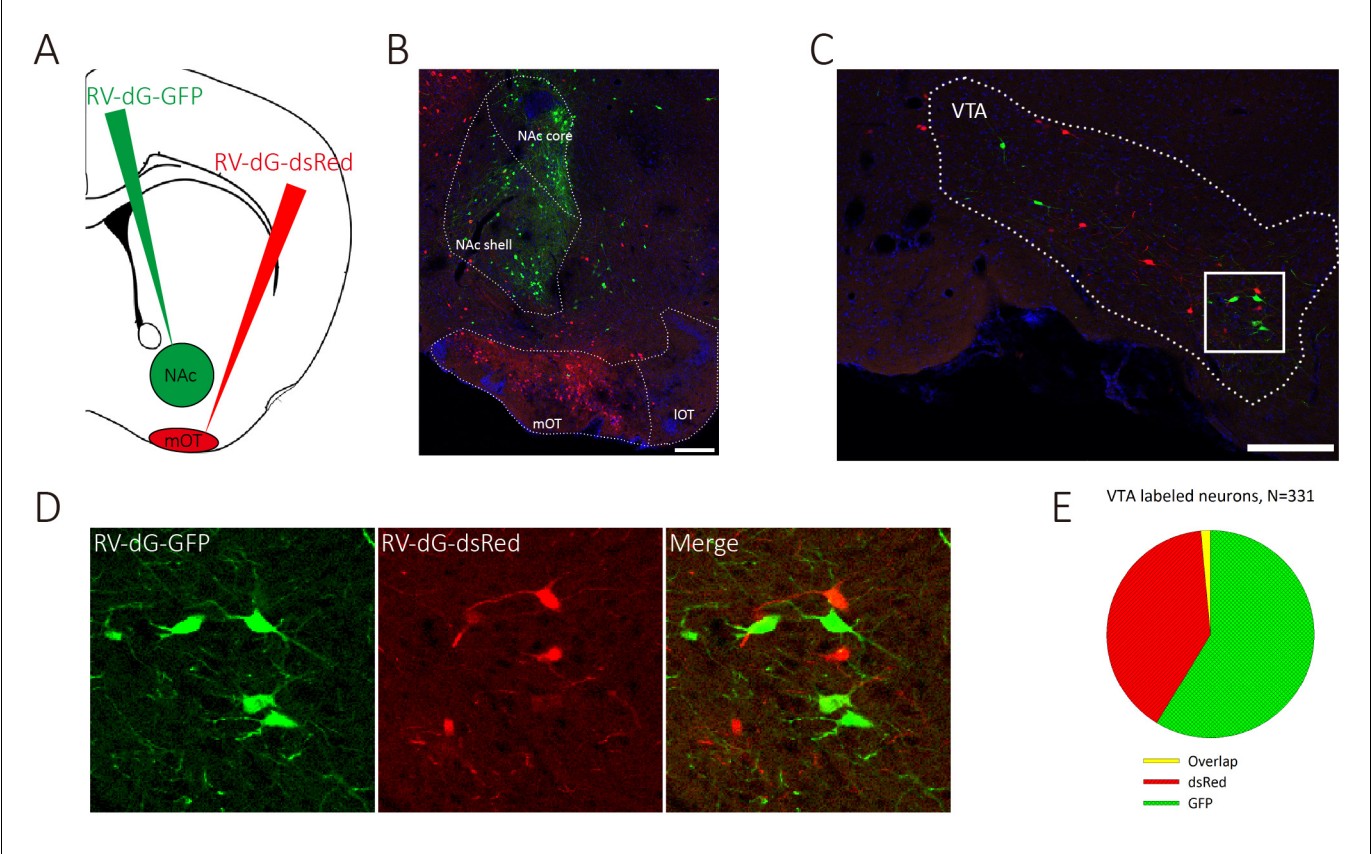

**Figure 6.** VTA DAergic neurons segregatedly innervate the NAc and the mOT. (**A**) Schematic for labeling the mOT and NAc co-innervated VTA neurons by dual RV labeling. (**B**) RV-dG-DsRed and RV-dG-GFP was injected into the mOT and NAc, respectively. (**C**) The VTA neurons were labeled by both RV-dG-DsRed and RV-dG-GFP. (**D**) The magnification of VTA-labeled neurons in C. (**E**) The co-labeled rate (1.65%) of the VTA neurons by both GFP and dsRed were negligible, which implied that it may not be common for a VTA DAergic neuron to co-innervate the NAc and the mOT. Scale bar: 200 μm.

DOI: https://doi.org/10.7554/eLife.25423.021

The following source data is available for figure 6:

**Source data 1.** Statistical result of percentage of investigation time for TMT after odor-light pairing for the ChR2-expressing mice and percentage of investigation time for geraniol after D1 and D2 dopamine receptor antagonist administration in the mOT.
DOI: https://doi.org/10.7554/eLife.25423.022

In the present study, we found that the VTA-mOT DAergic projection is activated by diverse naturalistic rewards, the activities of the mOT are altered by a wide range of odorants, and mice exhibited a preference for a given odor if that olfactory stimulant is paired with the activation of the VTA (DAergic)-mOT pathway. Our results thus provide an insight that the activation of the VTA-mOT DAergic projection by hedonic life experience might contribute to the generation of the preference for experience-related odors.

We showed that it is sufficient to artificially generate odor-preference by optogenetic activation of the VTA-mOT DAergic projection (*Figure 5*). Since this phenomenon is abolished by the dopamine receptor antagonist administration in the mOT, and it may not be common for a VTA DAergic neuron to co-innervate the NAc and the mOT (*Figure 6*), we can exclude the contributions of the activation of passing-by DAergic fibers in the mOT or VTA DAergic cell bodies, and confirm that the mOT dopamine released by the VTA-mOT DAergic projection should be responsible for this odor-preference formation. Besides, this pathway is also necessary for odor based reward learning and the expression of naturalistically formed odor-preference (*Figures 7* and *8*). However, whether or how this pathway plays roles in the formation of odor aversion, or the perception of the instinct preferred/aversive odors is still unknown.

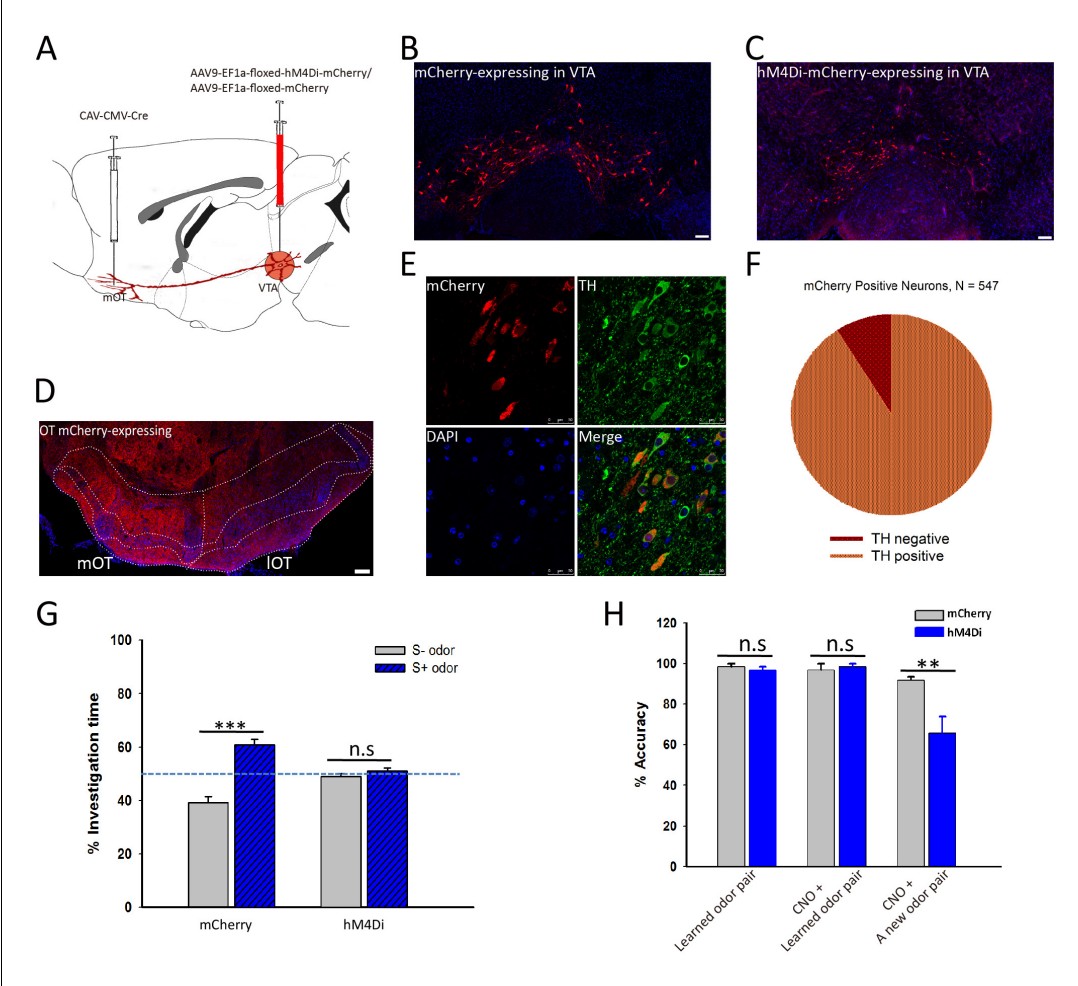

**Figure 7.** Inactivation of the mOT-projecting VTA DAergic neurons decreased the odor-cue based reward learning and learnt odor-preference. (**A**) The schematic for specifically targeting the mOT-projecting VTA DAergic neurons by injecting CAV-CMV-Cre and AAV-EF1a-floxed-hM4Di-mCherry/AAV-EF1a-floxed-mCherry into the mOT and VTA, respectively. (**B, C**) The mCherry-labeled neurons in the VTA in control and hM4Di-expressing mice. (**D**) The projections from the VTA mCherry-labeled neurons to the OT were mainly found in the medial part of the OT. (**E, F**) Most of the mCherry-labeled neurons in the VTA were also co-expressing TH (90.86%). (**G**) After the odor-preference was established (left bar pair), inhibition of the mOT-projecting VTA DAergic neurons abolished the paired preference (p=0.214, n = 5 for mCherry and 6 for hM4Di group, respectively). (**H**) Inactivation of this pathway in well-trained mice (left bar pair) did not influence the learnt Go-no go task performance (middle bar pair; Control group, p=0.373; hM4Di group, p=0.312), but significantly decreased the performance in learning a new pair of odors with the same paradigm (right bar pair; p<0.01, n = 5 for mCherry and 6 for hM4Di group, respectively).

DOI: https://doi.org/10.7554/eLife.25423.023

The following source data and figure supplement are available for figure 7:

**Source data 1.** Statistical result of odor-preference test after inhibition of the mOT-projecting VTA DAergic neurons, accuracy of Go-no go learning for hM4Di and control groups in well-trained mice.
DOI: https://doi.org/10.7554/eLife.25423.025
**Figure supplement 1.** Only a small portion of the VTA DAergic neurons were labeled by CAV-CMV-Cre.
DOI: https://doi.org/10.7554/eLife.25423.024

In the present study, inactivating the mOT projecting VTA neurons did not affect the learnt Go-no go performance (*Figure 7G,H*), mice were both highly motivated and still remembered how to seek water rewards. The possible reasons are that, the limited population of DAergic neurons labeled by CAV-CMV-Cre from the mOT may lead to incomplete inhibition or alternatively, the specific VTA (DAergic)-mOT pathway might not be necessary for established reward-seeking behavior. Thus, the other neural pathways such as the different populations of VTA DAergic neurons

projecting to the NAc, dorsal striatum and other brain regions may compensate the motivation and expression of reward seeking (*Hu, 2016*).

Inactivating the mOT projecting VTA neurons did not affect reward seeking, but strongly affected the learning efficiency for a new odor pair. Dopamine has been widely reported to strengthen environment stimuli-encoded glutamate inputs in the NAc, thus facilitated environment cue-based reward learning (*Flagel et al., 2011*; *Britt et al., 2012*). However, the function of the VTA (DAergic)-mOT pathway in reward learning has been rarely studied. Here, we reported that the pathway plays important roles in odor cue-based reward learning, toward which mice are able to establish an odor-preference. Since inactivating the mOT projecting VTA neurons abolished the preference for S + odor, together, we summarized that the VTA (DAergic)-mOT pathway is both necessary for the expression of established odor preference and may influence the establishment of odor-preference.

The reasons why paired activation of the VTA-mOT DAergic projection generates odor-preference could be complex. Given that the mOT is directly innervated by mitral/tufted cells from the OB (*Figure 1*;[*Igarashi et al., 2012*; *Scott et al., 1980*]) and participates in odor perception (*Figure 2—figure supplement 2*; [(*Agustín-Pavón et al., 2014*; *Mooney et al., 1987*; *Carlson et al., 2014*]), and that the D1R + neurons in the mOT are involved in reward-related odor perception, the first possibility is that activation of the VTA-mOT DAergic projection might induce long-term potentiation (LTP) in mOT D1R + neurons, and enhance the paired odors related input connections with these neurons. Hence, future occurrences of the paired odors should result in an increased rate of activation of D1R + neurons in the mOT, even without artificial activation of the VTA-mOT DAergic projection, as is the case in the NAc (*Keeler et al., 2014*; *Gerfen and Surmeier, 2011*; *Hikida et al., 2010*). Another possibility involves a broader network. In the present study, we found that activating the VTA-mOT DAergic projection induced increased c-fos expression in both the NAc and PCX (*Figure 3A–F*, *Figure 3—figure supplement 1*). Because the NAc is involved in reward (*Hu, 2016*; *Carlezon and Thomas, 2009*) and the PCX is responsible for olfactory memory and odor perception (*Wilson and Sullivan, 2011*), it's likely that the simultaneous activation of these two and the other brain regions may refer the olfactory memory of the paired odors to reward, leading to altered odor-preference. Further studies with specific manipulation and recording of neurons in the mOT and PCX are needed to address the mechanism in detail.

Animals learn to associate diverse sensory cues, not only odors, with rewards depending on their life experience. The striatum plays an important role in experience-dependent reward learning behaviors (*Mahon et al., 2004*; *Wickens et al., 2003*). Blockade of the striatum D1R + neurons damages both the visual cue-based reward learning and cocaine-induced conditional place preference (*Yawata et al., 2012*; *Hikida et al., 2010*). Besides, the activation of striatum D1R is necessary for the generation of corticostriatal LTP that induced by concurrent phasic dopamine release and cortical stimulation (*Keeler et al., 2014*; *Mahon et al., 2004*). Visual, auditory and other sensory cues can usually stimulate the cortical activation. As a part of the ventral striatum, the OT neurons also respond to an auditory stimulus (*Wesson and Wilson, 2010*). It is pleasant for pet dogs to hear the voice of their owners and for lovers to hear the melodies of the songs they sang together. Whether the VTA-mOT pathway influences these auditory preferences is also an interesting question.

## The functional differences between the mOT and the adjacent brain regions

The OT is composed of the medial and lateral parts, which are functionally different from each other (*Murata et al., 2015*; *DiBenedictis et al., 2015*; *Agustín-Pavón et al., 2014*; *Ikemoto, 2003*). Thus, how the VTA DAergic projection to the lateral OT affects and responds to reward or aversion compared with the mOT still remains to be examined.

It should also be pointed out that, although we intended to reveal the functions of the VTA-mOT DAergic projection in odor-preference, the border for the mOT and the lateral OT could be potentially imprecisely divided due to limited knowledge about this. Besides, it is also a difficult problem to perfectly demark the OT from the ventral pallidum. In the present work, we divided the mOT from the lateral OT at the most gyrated part of the layer 2 as we did before (*Zhang et al., 2017*), and assigned the mOT with the region containing the superficially located Islands of Calleja and surrounded cortex-like compartments according to the previous reports (*Murata et al., 2015*; *Xiong and Wesson, 2016*). Besides, the layers of the OT were also divided using methods in Xiong's

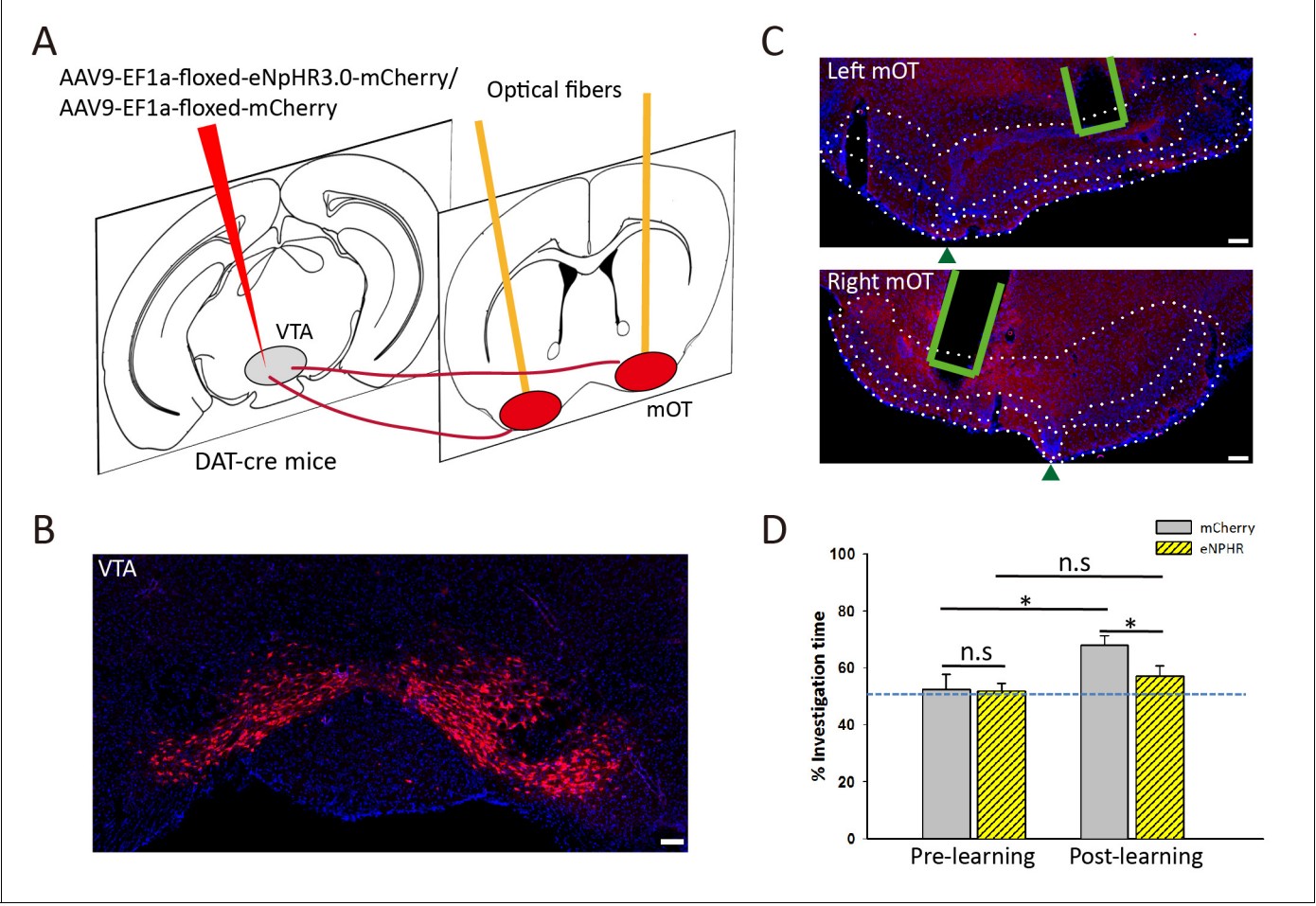

**Figure 8.** Optogenetic inactivation of VTA-mOT DAergic projections during odor-food associative learning abolished the establishment of S+ odor preference. (**A**) The schematic for viral injection and optic fiber implants. (**B**) eNpHR3.0-mCherry was densely expressed in the VTA neurons. (**C**) Representative coronal sections showed terminal expression of eNpHR3.0-mCherry and optic fiber implants in the bilateral mOT. The borders for the mOT and lateral OT were indicated by dark green triangles. Scale bar: 100 μm. (**D**) The S+ odor preference can't be established after odor-food associative learning by simultaneously light inhibition of the mOT DAergic projections from the VTA (each group n = 6). Scale bar: 100 μm. Figure supplement legends.

DOI: https://doi.org/10.7554/eLife.25423.026

The following source data is available for figure 8:

**Source data 1.** Statistical result of the odor-preference test after light inhibition of the mOT DAergic projections from the VTA.
DOI: https://doi.org/10.7554/eLife.25423.027

work (*Xiong and Wesson, 2016*). The border of the layer 3 was demarked by the deep layer's islands of Calleja.

Besides, the NAc and OT, components of the ventral striatum, mediate reward processing similarly (*Striano et al., 2014*; *Millhouse and Heimer, 1984*) and both are innervated by DAergic projections from the VTA. Although the NAc has been extensively studied, the OT has received much less attentions, in spite of the unique anatomical position. Previous studies found that the mOT robustly mediates self-administration of cocaine or D-amphetamine more than the NAc (*Ikemoto, 2005*; *Ikemoto, 2003*). Given that pharmacological agents can diffuse easily, it is difficult to specifically target the mOT while leaving the adjacent brain areas (like the NAc) undisturbed. Meanwhile, a recent research reported that optogenetic activation of DAergic terminals innervating the NAc is sufficient to drive-positive reinforcement (*Steinberg et al., 2014*), whether there would be any functional difference between activation of DAergic projection in the mOT and NAc is unknown.

Further studies using optogenetic activation of VTA DAergic projections in the NAc might be of great help to reveal their features.

In summary, we have provided evidence that the VTA-mOT DAergic projection can be activated by a variety of rewards and that activation of the projection can generate robust rewarding, leading to formation of olfactory as well as spatial preferences if the corresponding stimulation is coupled with the activation of the projection simultaneously. These results may underlie a circuit mechanism for how preference is influenced by life experience.

## Materials and methods

### Animals

All surgical and experimental procedures were conducted in accordance with the guidelines of the Animal Care and Use Committees at the Wuhan Institute of Physics and Mathematics, the Chinese Academy of Sciences (reference number: WIPM-(2014)39). Adult male and female C57BL/6 mice were purchased from Hunan SJA Laboratory Animal Company. DAT-Cre mice used were heterozygote and bred from DAT-internal ribosome entry site (IRES)-Cre (Jackson Laboratories, B6.SJL-Slc6a3tm1.1 (Cre) Bkmn/J, stock number 006660) (*Bäckman et al., 2006*) by mating the transgenic male mice with C57BL/6 females. All animals were housed with their littermates in a dedicated housing room with a 12/12 hr light/dark cycle. Food and water were freely available unless specifically noted.

### Viral constructs

For AAV viruses, the AAV9-EF1a-floxed-ChR2-mCherry, AAV9-EF1a-floxed-eNpHR3.0-mCherry, AAV9-EF1a-floxed-hM4Di-mCherry, AAV9-hsyn-Gcamp6s, and AAV9-EF1a-floxed-mCherry were packaged by BrainVTA Co., Ltd., Wuhan. AAV9-EF1a-DIO-GCaMP6m and AAV9-EF1a-DIO-EmGFP were gifts from Minmin Luo and constructed by replacing the coding region of ChR2-mCherry in the AAV9-EF1a-DIO-hChR2(H134R)-mCherry plasmid (constructed by K. Deisseroth) with those encoding enhanced membrane GFP (Addgene Plasmid 14757) or GCaMP6m (Addgene Plasmid 40754), respectively. All AAVs were purified and concentrated to titers of approximately $2 \times 10^{12}$ genome copies per milliliter.

For rabies viruses, the RV-dG-GFP and RV-dG-dsRed were packaged by standard methods as previous described (*Osakada et al., 2011*) with titers at about $1 \times 10^9$ pfu/ml.

To specifically inactivate the mOT-projecting VTA DAergic neurons, CAV-CMV-Cre was also packaged at titers of approximately $3 \times 10^{12}$ genome copies per milliliter by BrainVTA Co., Ltd., Wuhan.

### Stereotactic surgery

All procedures on animals were performed in Biosafety level 2 (BSL2) animal facilities as we did before (*Yang et al., 2016a*; *Yang et al., 2016b*). Briefly, animals were anesthetized with chloral hydrate (400 mg/kg, i.p.), and then placed in a stereotaxic apparatus (RWD, 68030). During surgery and virus injection, animals were kept anesthetized with isoflurane (1%). The skull above the targeted areas was thinned with a dental drill and removed carefully. Injections were conducted with a syringe pump (Quintessential stereotaxic injector, Stoelting, Cat#: 53311) connected to a glass micropipette with a tip diameter of 10–15 μm.

To retrograde trace the afferent neural circuit of the mOT, RV-dG-GFP (150 nL) was unilaterally injected into the mOT of the adult male C57BL/6 mice with the following coordinates: AP, 1.20 mm; ML, 1.10 mm; and DV, −5.50 mm. Seven days after virus injection, subjected mice were sacrificed.

To label the co-innervation of VTA neurons to the mOT and NAc, we co-injected 100 nL RV-dG-dsRed into the mOT and RV-dG-GFP into the NAc with the following coordinates: AP, 1.50 mm; ML, 0.7 mm; and DV, −4.50 mm. Seven days after virus injection, subjected mice were sacrificed.

To specifically manipulate and record the activities of the VTA-mOT DAergic projection, we respectively injected 300 nL of AAV9-EF1a-floxed-ChR2-mCherry, AAV9-EF1a-floxed-eNpHR3.0-mCherry or AAV9-EF1a-floxed-mCherry, AAV9-EF1a-DIO-GCaMP6m or AAV9-EF1a-DIO-EmGFP into the VTA of adult DAT-Cre mice with the following coordinates: AP, −3.20 mm; ML, 0.45 mm; and DV, −4.30 mm. Eight weeks later, subjects were ipsilaterally or bilaterally implanted with a chronic optical fiber (NA = 0.37, Φ = 200 μm; Fiblaser, Shanghai, China) or a cannula (OD = 0.48

mm, ID = 0.34 mm, RWD, 68030) targeted to the mOT with the following coordinates: AP, 1.20 mm; ML, 1.10 mm; and DV, −5.15 mm.

To specifically record the activities of the mOT neurons, we ipsilaterally injected 150 nL of AAV9-hsyn-Gcamp6s into the mOT of male C57BL/6J mice with the following coordinates: AP, 1.20 mm; ML, 1.10 mm; and DV, −5.50 mm. Two weeks later, subjects were implanted with a chronic optical fiber targeted to the mOT with the following coordinates: AP, 1.20 mm; ML, 1.10 mm; and DV, −5.15 mm. Implants were fixed to the skull using two small screws, Cyanoacrylate Adhesive (Tonsan Adhesive, Beijing, China) and followed by dental cement (New Century, Shanghai, China). After the surgery, lincomycin hydrochloride and lidocaine hydrochloride gel were applied to the incision sites to prevent inflammation and decrease pain. Mice were allowed to recover for at least 1 week before behavioral tests. Brain sections containing the mOT (AP from about 1.40 mm to 0.60 mm) were collected to confirm the site of optical fiber.

To specifically inactivate the mOT-projecting VTA DAergic neurons, 250 nL of CAV-CMV-Cre was bilaterally injected into the mOT of adult male C57BL/6 mice with the same coordinate as above, and 300 nL of AAV9-EF1a-floxed-hM4Di-mCherry or AAV9-EF1a-floxed-mCherry into the VTA of the C57BL/6 mice with the following coordinates: AP, −3.20 mm; ML, 0.45 mm; and DV, −4.30 mm. About 3 weeks later, subjects were implanted with a small home-made metal bark to the skull for head-fixed Go-no go training.

## Odor stimulation

For the odor stimulation, isoamyl acetate, carvone, citral, and geraniol (all from Sigma, MO) with 5% dilutions of saturated vapors were used. And mineral oil was used as a control. Each odor was presented for four consecutive trials. The duration of the odor pulse was 2 s with an inter-trial interval of 28 s to prevent habituation. Verification of the optic fiber placement were done at the end of the experiments, using methods described in our previous publication (*Li et al., 2011*).

## Optogenetic manipulations

### Slice recording

The DAT-Cre mice with AAV9-EF1α-DIO-ChR2-mCherry injected into the VTA were used. The mice were anesthetized with isoflurane. After rapid decapitation, the brains were quickly removed and placed into ice-cold oxygenated slicing solution saturated with mixture of 95% $O_2$ and 5% $CO_2$. Coronal sections (300 μm thick) of the VTA were acquired using a vibratome (Leica, VT1000S) at 0–4°C. The slices were incubated at 34°C for 30 min and then at room temperature (22–24°C) with oxygenated artificial cerebrospinal fluid (ACSF) consisted of (in mM): 118 NaCl, 2.5 KCl, 2 $CaCl_2$, 2 $MgCl_2$, 26 $NaHCO_3$, 0.9 $NaH_2PO_4$, 11 D-glucose. After incubation, a slice was transferred to the recording chamber, superfused with oxygenated ACSF at a rate of 2 ml/min for recording.

Whole cell recording was performed with borosilicate glass pipettes pulled to a tip resistance of 3 to 5 MΩ (Sutter Instrument Co., P-1000). Recording electrodes were filled with intracellular solution (in mM): 140 K-gluconate, 5 KCl, 2.5 $MgCl_2$, 10 HEPES, 4 Mg-ATP, 0.4 Na-GTP, 10 Na-phosphocreatine, 0.6 EGTA (pH 7.2, osmolarity 290 MOsm). Signals were recorded with a MultiClamp 700B amplifier (Molecular Devices, USA) and digitized at 10 kHz with DigiData 1440A (Molecular Devices, USA). The somas of ChR2-expressing VTA DAergic neurons (mCherry-positive) were easily identified. To optically stimulate ChR2-expressing neurons, a single-wavelength LED system (λ = 470 nm, CoolLED PE-100) with light pulses at 5 ms, 20 Hz was used. Data were analyzed using Clampfit 10.3 software and origin 9.0.

## Behavioral task

For all behavioral experiments, we handled the mice for 2 days before the training experiment, 3–5 min for each day, allowing them to get used to the tethering procedure. For optogenetic stimulation, the implanted optic fiber was connected to a blue light laser *via* patch cords (Fiblaser, Shanghai, China) and a fiber-optic rotary joint (Doric Lenses, Quebec, Canada), which could release the torsion of the fiber resulted from rotation of the mouse. All opto-stimulation experiments used 5 ms, 10–15 mW, 473 nm light pulses at 20 Hz via a solid-state laser for light delivery (Shanghai Laser and Optics Century, Shanghai, China) triggered by a stimulator (Model 2100, Isolated pulse stimulator, A-M systems).

## Real-time place preference (RTPP)

The mice were placed in a behavioral arena (50 × 50 × 25 cm) for 20 min and assigned one counter-balanced side of the chamber as the stimulation side. Mice were placed in the non-stimulation side at the onset of the experiment. Laser stimulation at 20 Hz were constantly delivered when they crossed to the stimulation side, but stopped when they returned back to initial non-stimulation side. All behavioral tests were recorded using an HD digital video camera (Sony, Shanghai, China) and the offline analyses were performed by a video tracking software (ANY-maze; Stoelting Co, IL).

## iClass

The behavioral arena (47 × 32 cm) contained a red rectangle as the target area for reinforcement, located in the center of the open field. The red central rectangle, formed by border line with the height of about 1 cm, is one-tenth of the whole arena. During the experiment, the floor of the arena was covered by fresh padding, with the red central area easily seen. The behavior task consisted of three phases (pre, light training, and test) and lasted for 5 consecutive days (*Liu et al., 2014*). During the pre-phase on day 1 (pre), mice with fiber patch cord connected were habituated in the arena without light pulses delivery. The basal locomotor activities of the mice were recorded and measured. The following light training sessions consisted of three consecutive days (L1-L3), on each day mice with fiber patch cord connected were put in the arena, and their activities were monitored for 15 min. When the centrals of the mice were located within the marked central rectangular area, blue light pulses at 20 Hz were constantly delivered into the mOT. Light stimulation was immediately terminated when the centrals of the mice left the red area. On test day (test), the same experimental procedures as in day 1 (pre) were performed.

## Odor-light pairing

The pairing procedure was performed for 2 consecutive days, which was two weeks post-surgery and plate placement. On each of the pairing days, the mouse was head restrained and adapted to the holding and odorant delivery conditions for 1–2 hr. Carvone and geraniol (Sigma, MO) were randomly delivered to the mouse's nose (2 s delivery, 30 s inter-stimulus interval, 20 trials each session for 5 sessions) with 5% of saturated vapors using a custom 8-channel olfactometer and balanced across mice. Only geraniol was delivered simultaneously with 2 s, 20 Hz trains of light pulses (473 nm). Control mice were treated identically. Furthermore, another pair of odors (v/v 5% TMT and sesame butter) was delivered in the same way, with TMT as the paired odor.

In the subsequent receptor blockade experiments, saline (200 nL in total) contained the D1R antagonist (SCH-23390, 1.0 mg/ml) and D2R antagonist (Raclopride, 1.5 mg/ml) as blocking agent, or saline alone (200 nL) as control, was administered into the mOT through the cannula 5 min before optogenetic activation of the VTA-mOT DAergic projection paired with odor stimulation, which were described as above.

## Odor-food reward associative learning

The protocols were adapted from Sophie's study (*Tronel and Sara, 2002*) with some modifications. Briefly, the investigation time of the mice to the two odors (geraniol and carvone) were measured (odor-preference test), and then mice were fasted 24 hr pre-training. The training apparatus was a square box with two sponges which had a hole of 1 cm in diameter cut into the center and were placed in glass side-holders of the same size. Two sponges were randomly placed in two opposite corners of the box. Food reinforcement was placed at the bottom of the opening in the sponge. Sponges were impregnated with an odor on each corner of the sponge. Training was performed in a single session with five trials. The mouse was introduced into the box at a corner without a sponge, head toward the wall. The spatial configuration of the sponges was changed between trials and the reinforcement was always associated with the geraniol. The constant 593 nm yellow light were constantly delivered into the mOT once the mouse was introduced into the box and during the training session. After the training was finished, the mice were taken to the odor-preference test.

## Odor-preference test

Mice were tested with the two odors (5% in mineral oil, v/v), applied by two glass sticks simultaneously, to their home cages. Home cage testing was chosen to minimize potential influences of

stress and anxiety (because of the new environment/context) on the behavioral measures. The food and water were removed from cages just before tests. Tests took place during the light phase of the animals' 12/12 hr dark/light cycle. Odors were delivered for 6 min by inserting the odorized sticks from one side of the animal's home cage. Mice were first habituated to the odor sticks for 1 min, and the time spent investigating, defined as sniffing within 1 cm of the odor stick, was recorded by a single observer blind to the experiment group over a 5-min period.

### Go-no go task

After about 24 hr of water deprivation, mice were exposed to these two odors randomly. They were trained to either lick the water pipe to get the reward (a drop of water) when they smelled one odor (defined as S+ odor), or wait without any licking when presented the other odor (defined as S- odor). After several days of training, when the accuracy reached above 85%, the mice were ready for further experiments. They were intraperitoneally injected with CNO (3 mg/kg of body weight) and then tested for odor-preference blockade experiments. Moreover, in the following days, the mice were trained to distinguish another pair of odors using the same principle they learned before, until the accuracy achieved above 85% again.

### Sucrose solution and food pellets consumption

Mice were water deprived for 24 hr before placed in a chamber (20 × 20 × 22 cm), which was equipped with a bottle filled with 5% (w/v) sucrose solution. The lick signals were recorded by a contact lickometer connected with a Power 1401 data acquisition system.

Mice were food deprived for 24 hr before placed in a chamber (20 × 20 × 22 cm). Food pellets were manually delivered to the chamber floor with only one pellet per time. Videos (10 min per session) from an overhead infrared camera were recorded. The onset of feeding was determined as the mouse picked up the pellet and started chewing.

### Social interaction

To study the effect of heterosexual interaction, the test male mouse was habituated in the dark with an optical fiber connected to the fiber ferrule on its head for 1 min. A female mouse was introduced into the home cage of the test male mouse. A recording session lasted for 10 min and the behavior of the test mouse was videotaped with an overhead infrared camera. In the male-male interaction sessions, a male mouse was introduced following the same procedure.

### Fiber photometry

The GCaMP fluorescent signals for sucrose or quinine solution, food pellets, social interactions and odors were recorded with the same setup (Fiber photometry, Thinker Tech Nanjing Biotech Limited Co.) as previously reported (Li et al., 2016).

### Immunohistochemistry

Briefly, 40-μm-thick coronal slices were obtained and the procedures for immunohistochemistry were performed similar to what we did before (Wei et al., 2015).

To determine the input patterns of the mOT, every sixth sections of the RV-labeled brain slices were collected and stained with DAPI. Four coronal slices from the VTA (AP from about −3.00 mm to −3.80 mm, spaced 240 μm from each other) per mouse were immunohistochemically stained for the tyrosine hydroxylase (TH, Abcam, Lot#: 2552365).

To determine the specificity of ChR2-mCherry expression in the VTA of DAT-Cre mice, 4 coronal slices from the VTA (AP from about −3.00 mm to −3.80 mm, spaced 240 μm from each other) per mouse (n = 3) were collected and immunohistochemically stained for the TH.

To detect the c-fos expression evoked by optical-manipulation, mice were put in an open field and received laser illumination for 10 min and were sacrificed 1.5 hr later. Every sixth brain sections (AP from about 1.40 mm to 0.50 mm) from three mice were collected and immunohistochemically stained for c-fos (Cell Signaling, #2250).

All the images were then captured and analyzed with TCS SP8 fluorescence laser scanning confocal microscope (Leica) and ImageJ software.

## Data analyses

For cell counting, the boundaries of brain regions were delineated manually with Photoshop based on the Allen Brain Atlas. The labeled neurons were quantified semi-automatically using FIJI and the cell counter plugin of ImageJ.

For statistical analyses, we first analyzed data from one mouse trial by trial, then calculated the mean value and standard errors of one group, then Independent-Samples T-Test were used to determine statistical differences between groups using SPSS (version 13.0). Statistical significance was set at ***$p < 0.001$; **$p < 0.01$; *$p < 0.05$. All data values were presented as mean ± s.e.m. Graphs were made using SigmaPlot (version 10.0).

## Acknowledgements

We thank Shanping Chen from Shenzhen Institutes of Advanced Technology for helping with slice recording, Yanqiu Li from Wuhan Institute of Physics and Mathematics for keeping and genotyping the DAT-Cre mice, Lingling Xu from Wuhan Institute of Physics and Mathematics for capturing confocal images, and Tung-Lin Wu from Vanderbilt University Medical Center (a native English speaker) to modify the manuscript. This work was supported financially by the National Basic Research Program (973 Program) of China (Grant No. 2015CB755600 to FX), Strategic Priority Research Program of Chinese Academy of Sciences (Grant No. XDB02050005 to FX), the National Natural Science Foundation of China (Grant No. 31771156, 81661148053 and 91632303 to FX and Grant No. 31400945 to QL, Grant No. 31400946 to XPR, Grant No. 31400977 to XBH).

## Additional information

### Funding

| Funder | Grant reference number | Author |
| --- | --- | --- |
| National Natural Science Foundation of China | 31400945 | Qing Liu |
| National Natural Science Foundation of China | 31400946 | Xiaoping Rao |
| National Natural Science Foundation of China | 31400977 | Xiaobin He |
| Ministry of Science and Technology of the People's Republic of China | 2015CB755600 | Fuqiang Xu |
| Chinese Academy of Sciences | XDB02050005 | Fuqiang Xu |
| National Natural Science Foundation of China | 81661148053 | Fuqiang Xu |
| National Natural Science Foundation of China | 91632303 | Fuqiang Xu |
| National Natural Science Foundation of China | 31771156 | Fuqiang Xu |

The funders had no role in study design, data collection and interpretation, or the decision to submit the work for publication.

### Author contributions

Zhijian Zhang, Conceptualization, Data curation, Formal analysis, Investigation, Writing—original draft, Project administration; Qing Liu, Data curation, Formal analysis, Funding acquisition, Investigation, Writing—review and editing; Pengjie Wen, Jiaozhen Zhang, Ziming Zhou, Juan Li, Xiaoran Xu, Xueyi Zhang, Rui Luo, Guanghui Lv, Pei Cao, Investigation, Writing—original draft; Xiaoping Rao, Data curation, Investigation; Hongruo Zhang, Investigation, Visualization; Xiaobin He, Resources, Funding acquisition, Project administration; Zheng Zhou, Resources, Writing—original draft;

Haohong Li, Liping Wang, Resources, Supervision; Fuqiang Xu, Conceptualization, Resources, Supervision, Funding acquisition, Writing—review and editing

### Author ORCIDs

Fuqiang Xu http://orcid.org/0000-0002-4382-9797

### Ethics

Animal experimentation: All surgical and experimental procedures were conducted in accordance with the guidelines of the Animal Care and Use Committee at the Wuhan Institute of Physics and Mathematics, the Chinese Academy of Sciences (reference number: WIPM-(2014)39). All Animals were housed with their littermates in a dedicated housing room with a 12/12 h light/dark cycle. Food and water were available free unless specifically noted. During the surgery, every effort was made to minimize suffering.

### Decision letter and Author response

Decision letter https://doi.org/10.7554/eLife.25423.030
Author response https://doi.org/10.7554/eLife.25423.031

## Additional files

### Supplementary files

• Transparent reporting form
DOI: https://doi.org/10.7554/eLife.25423.028

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
