## [Decision Letter]

Thank you for submitting your article "Activation of the Dopaminergic Pathway from VTA to the Medial Olfactory Tubercle Generates Odor-Preference and Reward" for consideration by *eLife*. Your article has been reviewed by three peer reviewers, and the evaluation has been overseen by a Reviewing Editor and a Senior Editor. The reviewers have opted to remain anonymous.

The reviewers have discussed the reviews with one another and the Reviewing Editor has drafted this decision to help you prepare a revised submission. All reviewers agree that showing that the DA-mOT pathway mediates reward processing and olfactory learning is indeed novel and interesting. However, the reviewers were also unanimous about the potential confounding factor of non-selective activation of DA projections to other brain areas, for example the NAc. Although the others provide some pharmacology to try to tackle this, the reviewers were not convinced by these data. The authors need to conduct some additional experiments to strengthen the study. The authors could demonstrate that the DA neurons separately project to the NAc and the mOT. If this turns out not to be the case, a combination of optogenetic stimulation of DA fibers in the mOT with pharmacological inactivation of DA cell bodies would be important. This would at least rule out the potential effect of antidromic stimulation, although the activation of axonal collaterals can still be an issue.

Additionally we attach the point-by-point comments of the reviewers below.

Reviewer #1:

The medial olfactory tubercle (mOT) receives dense dopaminergic inputs from the ventral tegmental area and has often been considered an extension of the nucleus accumbens shell (NAc) in the ventral striatum. Although the roles of the NAc in reward processing have been the major focus of numerous research groups, the mOT has also been implicated in drug rewards. Unlike the NAc, the mOT receives strong direct olfactory inputs. Wesson and Wilson (2011) put forward some intriguing hypotheses regarding the potential functions of the mOT in olfactory processing, but progresses thus far on this mysterious structure have been very limited.

Here, Zhang et al. from Fuqiang Xu's group provide several interesting observations indicating that the dopaminergic projection from the VTA to the mOT might mediate a variety of naturalistic reward processes and contributes to the formation of experience-dependent odor preference. They mapped the presynaptic partners of mOT neurons using rabies virus. By employing fiber photometry, they demonstrated that natural rewards activate dopaminergic terminals in the mOT. Optogenetically stimulating these terminals reinforces animal behavior and guides the formation of odor preference in a dopamine receptor-dependent manner. Finally, they showed that chemogenetically inactivating mOT-projecting neurons in the VTA abolishes conditioned odor preference and disrupts odor-discrimination learning using the go/no-go paradigm.

I like many aspects of the study. The authors took a comprehensive approach and used several latest tools for functional dissection of neural circuits. The data are overall solid. The conclusions should be exciting to a broad audience working on reward-related behaviors and olfactory processing. I am mainly concerned that some of the behavioral effects might be produced by the projection of VTA dopamine neurons to the NAc through axonal collaterals or fibers of passage. The authors performed some experiments to address this potential caveat. For example, infusing dopamine blockers into the mOT and chemogenetically inhibiting mOT-projecting VTA neurons significantly reduce the behavioral effect of optogenetic stimulation. However, drugs may diffuse into the nearby NAc region. Similarly, injecting canine adenovirus (CAV) into the mOT might have labeled VTA neurons projecting to the NAc.

The authors have several choices to address this concern. First, they could examine whether dopamine neurons projecting to the NAc and the mOT belong to separate subpopulations. Second, they could test whether the reinforcing effect of optogenetic stimulation in the mOT remains intact following lidocaine administration into the VTA. Third, they could show whether CAV-Cre injection into the mOT only drives gene expression in mOT-projecting dopamine neurons from the VTA. To the minimum, if none of these experiments work out, the authors should discuss the potential caveat and tone down some of the conclusions.

Reviewer #2:

This is a potentially interesting manuscript proposing that the dopaminergic innervation of the medial olfactory tubercle (mOT) by dopaminergic neurons from the ventral tegmental area (VTA) plays a role in reward. The authors show projection of dopaminergic neurons from the ventral tegmental area (VTA) to the mOT. They show that the neuronal circuit of mOT responds to reward and to odorants. They then proceed to activate or inhibit dopaminergic axons in the mOT and they argue that these manipulations elicit changes in behavior consistent with a role for mOT in reward. Unfortunately, my enthusiasm is diminished significantly by substantial problems outlined below.

1) A potential major problem is that optogenetic stimulation might have resulted in direct or indirect stimulation of the nucleus accumbens (NAcc). The authors should provide evidence that the behavioral results for optogenetic stimulation of mOT are not due to direct stimulation of NAcc. After all c-Fos was increased in NAcc (Figure 5).

2) The viral injections targeting mOT might have resulted in infection of neurons in NAcc. The figures need to be modified to clearly show that this is not the case. Was this tested for every single experiment? This criticism holds for Figure 1 and 2.

3) For Figure 1 the authors should include an image showing that the infection did not spread to NAcc. What is the blue label? Was the slice in Figure 1 a coronal slice? If that is the case why is the lesion caused by the syringe circular in this plane? Are the cells surrounding the incision site neurons, or glia? They do not have the typical morphology of neurons. Perhaps the authors should show a slice that does not include the lesion. Please include a panel at a high magnification of the olfactory bulb showing clearly whether tufted cells were labeled.

4) The LFP is an indirect measure of activity in the mOT. Did the authors record odorant-elicited changes in single unit spikes or neuronal activity measured with photometry? Why is the power increasing before the odorant is presented?

5) The panel in Figure 4 shows a large number of mCherry cells that are not TH positive. This is discordant with the pie chart in Figure 4.

6) The authors claim that the experiment in Figure 9 shows that inactivation of DAergic neurons decreased the learned odor preference. But the viral infection resulted in expression of mCherry in a large number of non-DA neurons. Could the effect be due to GABAergic neurons projecting from VTA to mOT? Did the authors perform this experiment with halorhodopsin?

7) The statistics of the manuscript are not acceptable. Which statistical test was used for Figure 2,Figure 5,Figure 6,Figure 7,Figure 8 and 9? Please provide a statistical analysis of significance for Figure 7

8) For the experiment in Figure 2 are the neurons in mOT responding to all stimuli as opposed to an exclusive response to reward? Did the authors do a control where the mice were presented with a stimulus that is not a reward (bitter food pellets)?

Reviewer #3:

In this manuscript the authors reported experiments aiming to address the role of the dopaminergic (DAergic) innervation of medial olfactory tubercle (mOT) from the ventral tegmental area (VTA) in the regulation of olfaction and reward. Olfactory tubercle has been implicated in both reward and olfactory information processing. However, in both the olfactory field and in the reward field, OT is a far less studied brain region compared with other olfactory regions and reward regions. The role of the VTA(DAergic)-mOT pathway in the regulation of reward and olfactory process have not been investigated in behaving animals. This work represents an important and timely study addressing fundamental questions regarding the circuit mechanisms underlying the behaviors involving olfactory processing and/or reward. With the presynaptic optogenetic activation, local administration of DA receptor antagonists in mOT, and chemogenetic silencing of mOT-projecting VTA DAergic neurons, the importance of the DAergic input from VTA to mOT in the odor-preference is reasonably well established. However, the role of this pathway in other reward processes is far less clear with the current experiments.

The major issue of this study concerns the broad and non-specific effect of the optogenetic and chemogenetic manipulations. VTA DAergic neurons innervate multiple regions of the brain, and the neurons that innervate mOT likely also innervate other brain regions. Indeed, optogenetic activation of the VTA-mOT DAergic terminals was able to activate multiple brain regions (Figure 5). For optogenetic manipulations, the authors certainly realized this issue, as they have clearly stated, "[…] optical stimulation of the VTA-mOT DAergic projection might lead to the activation of the passing-by DAergic fibers in the mOT or VTA DAergic cell bodies, which may then activate afferents projecting to the other brain regions." The authors addressed this issue in one odor preference test by specifically administering DA receptor antagonists to mOT, and the results showed convincingly that the DA input from VTA to mOT is necessary for the odor-preference. However, the similar antagonist administration experiments were not conducted in the place preference behavioral assays (Figure 6 and Figure 7). Therefore, it is possible that the reward behaviors elicited by optogenetic activation of the VTA-mOT DAergic terminals may result from non-specific activation of other brain regions innervated by the activated VTA DAergic neurons, instead of activation of the VTA(DAergic)-mOT pathway. The authors should either tone down their conclusion to reflect this possibility or conduct at least the mOT DA receptor antagonist experiments to validate their conclusion regarding other preferences.

In the case of the chemogenetic silencing experiments (Figure 9), the global inhibition of the CAV-Cre/AAV9-EF1a-floxed-hM4Di-mCherry infected VTA neurons would also abolish their function in other brain regions. Therefore, it is possible that the attenuation in the odor-cue based reward learning (Figure 9H) may result from non-specific silencing of the projections to other brain regions. It is not clear what portion of VTA DAergic neurons are labeled by CAV-Cre/AAV9-EF1a-floxed-hM4Di-mCherry. If there were only a small portion labeled, it could be less problematic. Also, could the authors administer CNO locally at mOT to achieve local inhibition of VTA-mOT terminals?

[Editors' note: further revisions were requested prior to acceptance, as described below.]

Thank you for submitting your article "Activation of the Dopaminergic Pathway from VTA to the Medial Olfactory Tubercle Generates Odor-Preference and Reward" for consideration by *eLife*. Your article has been reviewed by three peer reviewers, and the evaluation has been overseen by a Reviewing Editor and a Senior Editor. The following individuals involved in review of your submission have agreed to reveal their identity: Minmin Luo (Reviewer #1); Diego Restrepo (Reviewer #2).

The reviewers have discussed the reviews with one another and the Reviewing Editor has drafted this decision to help you prepare a revised submission.

The reviewers agree that the main concerns have been mitigated by the new experiments and there are only text corrections at this point. The authors should provide detailed parameters for the lidocaine experiment (Figure 4).

In addition, the experiments that rule out co-lateralization should be a main figure. There are other small suggestions that the authors can address.

Reviewer #1:

Zhang et al. investigated the role of the dopaminergic projection from the ventral tegmental area (VTA) to the medial olfactory tubercle (mOT) in reward processing. Previously I raised the major concern that some of the behavior effects induced by optogenetic stimulation might be produced by collaterally activating dopaminergic fibers in the nucleus accumbens (NAc). In the revision, the authors performed several new experiments to address this and my other concerns. They now show that the NAc and the mOT receive largely separate dopaminergic inputs. In addition, optogenetically inhibiting dopaminergic activity in the mOT blocks many of the reward-like effects. These new results make the authors' conclusion much more convincing. The presentation of the manuscript has also been substantially improved.

Reviewer #2:

Outstanding; the authors have addressed the questions thoroughly.

Reviewer #3:

In the revised manuscript the authors conducted two sets of experiments, as per reviewer 1's suggestion, to address the major concern raised by all reviewers concerning the potential non-specific effect of the optogenetic and chemogenetic manipulations. In one, the authors conducted dual tracing experiments by injecting in the same animal distinct retrograde labeling viral vectors to mOT and NAc, respectively. They found that there was only a very small percentage (< 2%) of VTA neurons being double labeled (Figure supplement 4, C-G). The data suggest that VTA comprises separate populations of neurons innervating mOT and NAc. This is a significant finding. The result provides a strong support for their interpretation about the observed behavioral outcome being the specific effect of manipulation of VTA-mOT pathway. With this piece of data, the major concern on the previous submission of this manuscript should be satisfactorily addressed. I would suggest including this piece of data in the main body of paper, not as a figure supplement.

In another set of experiments, authors conducted real-time place preference test upon VTA administration of lidocaine and optogenetic stimulation of VTA-mOT projecting terminals (Figure 4). The authors showed that there was still a low level of place preference. However, there are several issues involved in this experiment. First, there is no information provided in the manuscript regarding the dose of drug, the efficiency of inhibition, the total distance of movement after inhibition etc. The small enhancement of preference in the stimulation side could be due to incomplete inhibition of VTA. Second, the effect of global inhibition of VTA on specific reward behaviors is not known, and a control for non-stimulation is lacking. Third, the data analysis can be problematic. The authors made the comparison between the two sides of the chamber, not the stimulated-side to random. To me, this experiment in its current stage does not add much to this paper. I would suggest the authors not to include this piece of data in the manuscript.

In addition, Figure 7 shows a nice and interesting result from a new experiment of optogenetic inhibition. It should not be included in Figure 7. Instead, it is better to combine Figure 7I and Supplemental Figure 5 and to present them as a new figure in the main text.

In the figure legend for Figure supplement 4, "(F) The magnification of VTA labeled neurons in B." should be "(F) The magnification of VTA labeled neurons in E".

---

## [Author Response]

Reviewer #1:[…] The authors have several choices to address this concern. First, they could examine whether dopamine neurons projecting to the NAc and the mOT belong to separate subpopulations. Second, they could test whether the reinforcing effect of optogenetic stimulation in the mOT remains intact following lidocaine administration into the VTA. Third, they could show whether CAV-Cre injection into the mOT only drives gene expression in mOT-projecting dopamine neurons from the VTA. To the minimum, if none of these experiments work out, the authors should discuss the potential caveat and tone down some of the conclusions.

We deeply appreciate your pertinent concerns and constructive suggestions. We have attempted to follow your suggestions closely to revise the manuscript.

To exclude the potential caveat, we have done a series of new experiments you have suggested (2 of the 3) and an extra one not suggested.

1) Virus tracing experiment. We co-injected RV-dG-dsRed into the mOT and RV-dG-GFP into the NAc, respectively, and found that the co-labeled rate of the VTA neurons were negligible (1.65%), which implied that it may not be common for a VTA DAergic neuron to innervate both the NAc and the mOT (Figure 6). However, limited by the efficiency of RV infection, the mOT-projecting VTA DAergic neurons may hardly be completely labeled. Thus we have also reminded readers about the potential caveat in the Discussion section (subsection “VTA-mOT DAergic Projection in Reward”).

2) We have conducted additional experiments on pharmacological inhibition of the VTA cell bodies, followed by optogenetic stimulation of the VTA-mOT DAergic projection. First, we found that the acute place preference induced by activation of the VTA-mOT DAergic projection were still retained even though the VTA cell bodies were inhibited by lidocaine (Figure 4), suggesting that the VTA-mOT DAergic pathway is involved in place preference formation. Second, after odor-preference was established, inhibition of the mOT- projecting VTA DAergic neurons (through hM4Di) abolished the paired odor preference (Figure 7), and inactivation of this pathway significantly decreased the performance in learning a new pair of odors in well-trained mice for Go-no go task (Figure 7). In summary, these experimental data provided strong evidence for the necessity and adequate roles of the pathway in preference formation.

Reviewer #2:[...] 1) A potential major problem is that optogenetic stimulation might have resulted in direct or indirect stimulation of the nucleus accumbens (NAcc). The authors should provide evidence that the behavioral results for optogenetic stimulation of mOT are not due to direct stimulation of NAcc. After all c-Fos was increased in NAcc (Figure 5).

We agree with this suggestion, therefore we have performed a series of new experiments to exclude this possibility, from three different aspects (that is, anatomical connections, local temporary anesthesia, and long-term chemogenetic manipulation). The detailed information is presented above (Reply for major comments of reviewer #1).

2) The viral injections targeting mOT might have resulted in infection of neurons in NAcc. The figures need to be modified to clearly show that this is not the case. Was this tested for every single experiment? This criticism holds for Figure 1, Figure 2

Thanks for the reviewer's suggestion. We have re-done the injection experiments and modified the figures. The injection site for every mouse was examined at the end of the experiments. The results are shown in Figure supplement 4 in the revised version.

3) For Figure 1 the authors should include an image showing that the infection did not spread to NAcc. What is the blue label? Was the slice in Figure 1 a coronal slice? If that is the case why is the lesion caused by the syringe circular in this plane? Are the cells surrounding the incision site neurons, or glia? They do not have the typical morphology of neurons. Perhaps the authors should show a slice that does not include the lesion. Please include a panel at a high magnification of the olfactory bulb showing clearly whether tufted cells were labeled.

We deeply agree with the reviewer’s suggestion. We have re-done the injecting experiments and replaced Figure 1 with new images. Cell nuclei in different coronal slices were stained in blue color by DAPI. High magnifications showed that cells labeled in the OB were mainly distributed in the mitral cell layers, while also sparsely located in the external plexiform layers (Figure 1).

4) The LFP is an indirect measure of activity in the mOT. Did the authors record odorant-elicited changes in single unit spikes or neuronal activity measured with photometry? Why is the power increasing before the odorant is presented?

We agree with the reviewer's concern. Therefore, we have done new experiments to record odorant-elicited neuronal activity with fiber photometry. We have added this result in the revised version in Figure 2—figure supplement 2 as a replacement for the original Figure 3.

5) The panel in Figure 4 shows a large number of mCherry cells that are not TH positive. This is discordant with the pie chart in Figure 4.

We agree with the reviewer. This discordant may be caused by the bad image quality. We have counted more cells, and drawn a new pie chart accordingly. We have performed more immunostaining experiments and obtained images with much improved quality. The new results have been presented in Figure 3 of the revised version.

6) The authors claim that the experiment in Figure 9 shows that inactivation of DAergic neurons decreased the learned odor preference. But the viral infection resulted in expression of mCherry in a large number of non-DA neurons. Could the effect be due to GABAergic neurons projecting from VTA to mOT? Did the authors perform this experiment with halorhodopsin?

We have re-done the staining experiment with a higher staining quality and acquired the image with higher resolutions. It turned out that most of the mCherry+ neurons are also TH positive. We pulled the data together and found that about 90.86% (vs 71.14% in the original version) mCherry labeled neurons were dopaminergic. Moreover, we have performed an inactivation experiment by injecting AAV-DIO-eNpHR in DAT-cre mice, and performed an inhibition experiment with halorhodopsin and found that the S+ odor preference can’t be established after odor- food associative learning. These data has been added to the revised version (Figure 7).

7) The statistics of the manuscript are not acceptable. Which statistical test was used for Figure 2,Figure 5,Figure 6,Figure 7,Figure 8 and 9? Please provide a statistical analysis of significance for Figure 7.

We agree with the suggestion, and have added the specific statistical tests for Figure 2,Figure 5,Figure 6,Figure 7,Figure 8 and 9. However, the relationships within the original Figure 7 (new Figure 5) are too complicated. It is difficult to present the significances of the same inter-groups with different time points, or the significances of the same time points with different inter-groups. Besides, although we have the data, the significant differences within this figure are not important as the focus of this study. Therefore, we have not provided a statistical analysis of significance for Figure 7. Anyway, we can add this information if you think it is necessary.

8) For the experiment in Figure 2 are the neurons in mOT responding to all stimuli as opposed to an exclusive response to reward? Did the authors do a control where the mice were presented with a stimulus that is not a reward (bitter food pellets)?

Thanks for the helpful suggestion. As shown in Figure 2, the neurons in the mOT responded to all reward stimuli we applied, such as food, water and social stimulation. We have added a new experiment with fiber photometry to record the mOT response of mice presented with quinine or sucrose solution. We found that the mOT neurons also responded to bitter water though the amplitude was much smaller than the sucrose solution (Figure 2—figure supplement 1).

Reviewer #3:In this manuscript the authors reported experiments aiming to address the role of the dopaminergic (DAergic) innervation of medial olfactory tubercle (mOT) from the ventral tegmental area (VTA) in the regulation of olfaction and reward. Olfactory tubercle has been implicated in both reward and olfactory information processing. However, in both the olfactory field and in the reward field, OT is a far less studied brain region compared with other olfactory regions and reward regions. The role of the VTA(DAergic)-mOT pathway in the regulation of reward and olfactory process have not been investigated in behaving animals. This work represents an important and timely study addressing fundamental questions regarding the circuit mechanisms underlying the behaviors involving olfactory processing and/or reward. With the presynaptic optogenetic activation, local administration of DA receptor antagonists in mOT, and chemogenetic silencing of mOT-projecting VTA DAergic neurons, the importance of the DAergic input from VTA to mOT in the odor-preference is reasonably well established. However, the role of this pathway in other reward processes is far less clear with the current experiments.

We deeply appreciate your valuable comments and helpful suggestions. We have attempted to follow your suggestions closely to revise the manuscript. The focal point of this study was that the VTA-mOT DAergic projection plays important roles in the formation of odor-preference and place/location preference. The role of this pathway in other reward processes should be an interesting project for the future.

The major issue of this study concerns the broad and non-specific effect of the optogenetic and chemogenetic manipulations. VTA DAergic neurons innervate multiple regions of the brain, and the neurons that innervate mOT likely also innervate other brain regions. Indeed, optogenetic activation of the VTA-mOT DAergic terminals was able to activate multiple brain regions (Figure 5). For optogenetic manipulations, the authors certainly realized this issue, as they have clearly stated, "… optical stimulation of the VTA-mOT DAergic projection might lead to the activation of the passing-by DAergic fibers in the mOT or VTA DAergic cell bodies, which may then activate afferents projecting to the other brain regions." The authors addressed this issue in one odor preference test by specifically administering DA receptor antagonists to mOT, and the results showed convincingly that the DA input from VTA to mOT is necessary for the odor-preference. However, the similar antagonist administration experiments were not conducted in the place preference behavioral assays (Figure 6 and Figure 7). Therefore, it is possible that the reward behaviors elicited by optogenetic activation of the VTA-mOT DAergic terminals may result from non-specific activation of other brain regions innervated by the activated VTA DAergic neurons, instead of activation of the VTA(DAergic)-mOT pathway. The authors should either tone down their conclusion to reflect this possibility or conduct at least the mOT DA receptor antagonist experiments to validate their conclusion regarding other preferences.

Thanks for the reviewer's suggestion. However, the antagonist may easily diffuse into the nearby NAc region through mOT infusing, as the other reviewers have pointed out. So, we have conducted the optogenetic RTPP tests when the VTA cell bodies were inhibited by lidocaine. We found that activation of the VTA (DAergic)- mOT pathway produced place preference even though the VTA cell bodies were inhibited (Figure 4).

In the case of the chemogenetic silencing experiments (Figure 9), the global inhibition of the CAV-Cre/AAV9-EF1a-floxed-hM4Di-mCherry infected VTA neurons would also abolish their function in other brain regions. Therefore, it is possible that the attenuation in the odor-cue based reward learning (Figure 9H) may result from non-specific silencing of the projections to other brain regions. It is not clear what portion of VTA DAergic neurons are labeled by CAV-Cre/AAV9-EF1a-floxed-hM4Di-mCherry. If there were only a small portion labeled, it could be less problematic. Also, could the authors administer CNO locally at mOT to achieve local inhibition of VTA-mOT terminals?

Thanks for the reviewer’s suggestion. The problem you are concerned with is similar to reviewers #1&2. We have performed a series of new experiments to exclude this possibility. The detailed information is presented above (please see Reply for major comment of reviewer #1). We have counted the infected VTA DA neurons and the whole VTA DA neurons, and found that there were only a small portion of the VTA DA neurons projecting to the mOT (37.20%), which should make the “axonal collaterals” less problematic. As for the local administration of CNO at mOT, a paper showed in Nature Neuroscience 17, 577–585 (2014) that it could be used to manipulate axonal terminals of DREADD-expressing neurons, we believe that local administration of CNO at mOT could inhibit the VTA- mOT terminals. However, CNO may also diffuse, as the same with dopamine blockers, into the nearby NAc region.

[Editors' note: further revisions were requested prior to acceptance, as described below.]

[…] Reviewer #3:In the revised manuscript the authors conducted two sets of experiments, as per reviewer 1's suggestion, to address the major concern raised by all reviewers concerning the potential non-specific effect of the optogenetic and chemogenetic manipulations. In one, the authors conducted dual tracing experiments by injecting in the same animal distinct retrograde labeling viral vectors to mOT and NAc, respectively. They found that there was only a very small percentage (< 2%) of VTA neurons being double labeled (Figure supplement 4, C-G). The data suggest that VTA comprises separate populations of neurons innervating mOT and NAc. This is a significant finding. The result provides a strong support for their interpretation about the observed behavioral outcome being the specific effect of manipulation of VTA-mOT pathway. With this piece of data, the major concern on the previous submission of this manuscript should be satisfactorily addressed. I would suggest including this piece of data in the main body of paper, not as a figure supplement.

We deeply appreciate and agree with your positive comments. We have transferred the original Figure supplement 4C-G into the main body of the revised manuscript as new Figure 6.

In another set of experiments, authors conducted real-time place preference test upon VTA administration of lidocaine and optogenetic stimulation of VTA-mOT projecting terminals (Figure 4). The authors showed that there was still a low level of place preference. However, there are several issues involved in this experiment. First, there is no information provided in the manuscript regarding the dose of drug, the efficiency of inhibition, the total distance of movement after inhibition etc. The small enhancement of preference in the stimulation side could be due to incomplete inhibition of VTA. Second, the effect of global inhibition of VTA on specific reward behaviors is not known, and a control for non-stimulation is lacking. Third, the data analysis can be problematic. The authors made the comparison between the two sides of the chamber, not the stimulated-side to random. To me, this experiment in its current stage does not add much to this paper. I would suggest the authors not to include this piece of data in the manuscript.

We deeply appreciate your constructive suggestions, and have deleted this piece of data and text accordingly in the revised version.

In addition, Figure 7 shows a nice and interesting result from a new experiment of optogenetic inhibition. It should not be included in Figure 7. Instead, it is better to combine Figure 7 and Supplemental Figure 5 and to present them as a new figure in the main text.

We agree with this helpful suggestion. We have combined the original Figure 7 and Supplemental Figure 5 in the revised version as new Figure 8, and have changed the relevant context accordingly.

In the figure legend for Figure supplement 4, "(F) The magnification of VTA labeled neurons in B." should be "(F) The magnification of VTA labeled neurons in E".

Thank you so much for your careful reading, we have revised it in the revised version.